# Epistasis facilitates functional evolution in an ancient transcription factor

Brian PH Metzger[1†], Yeonwoo Park[2‡], Tyler N Starr[3§], Joseph W Thornton[1,4*]

[1]Department of Ecology and Evolution, University of Chicago, Chicago, United States; [2]Program in Genetics, Genomics, and Systems Biology, University of Chicago, Chicago, United States; [3]Department of Biochemistry and Molecular Biophysics, University of Chicago, Chicago, United States; [4]Department of Human Genetics, University of Chicago, Chicago, United States

**\*For correspondence:** joet1@uchicago.edu

**Present address:** [†]Department of Biological Sciences, Purdue University, West Lafayette, United States; [‡]Center for RNA Research, Seoul National University, Seoul, South Korea; [§]Department of Biochemistry, University of Utah, Salt Lake City, United States

**Competing interest:** The authors declare that no competing interests exist.

**Abstract** A protein's genetic architecture – the set of causal rules by which its sequence produces its functions – also determines its possible evolutionary trajectories. Prior research has proposed that the genetic architecture of proteins is very complex, with pervasive epistatic interactions that constrain evolution and make function difficult to predict from sequence. Most of this work has analyzed only the direct paths between two proteins of interest – excluding the vast majority of possible genotypes and evolutionary trajectories – and has considered only a single protein function, leaving unaddressed the genetic architecture of functional specificity and its impact on the evolution of new functions. Here, we develop a new method based on ordinal logistic regression to directly characterize the global genetic determinants of multiple protein functions from 20-state combinatorial deep mutational scanning (DMS) experiments. We use it to dissect the genetic architecture and evolution of a transcription factor's specificity for DNA, using data from a combinatorial DMS of an ancient steroid hormone receptor's capacity to activate transcription from two biologically relevant DNA elements. We show that the genetic architecture of DNA recognition consists of a dense set of main and pairwise effects that involve virtually every possible amino acid state in the protein-DNA interface, but higher-order epistasis plays only a tiny role. Pairwise interactions enlarge the set of functional sequences and are the primary determinants of specificity for different DNA elements. They also massively expand the number of opportunities for single-residue mutations to switch specificity from one DNA target to another. By bringing variants with different functions close together in sequence space, pairwise epistasis therefore facilitates rather than constrains the evolution of new functions.

## eLife assessment

This study includes **fundamental** findings on protein evolution, namely that changes in function are largely attributable to pairwise rather than higher-order interactions, and that epistasis potentiates rather than constrains evolutionary paths. **Compelling** evidence supporting the conclusions is provided by applying a new model to a previously generated experimental dataset on deep mutational scanning of the DNA-binding domain (DBD) of steroid hormone receptor. The implications of this work are of considerable interest to protein biochemistry, evolutionary biology, and numerous other fields.

## Introduction

A protein's genetic architecture – the set of causal rules by which its sequence determines its functions – is of fundamental importance in genetics, biochemistry, and evolution. These rules include the effect

on function of every possible amino acid state at each site in the protein, and the epistatic effects of all possible pairs and higher-order combinations. This genetic architecture determines the distribution of functions across the space of all possible protein sequences and, in turn, the paths accessible to an evolving protein under the influence of mutation, drift, and selection.

The extent and impact of epistatic interactions is of particular interest. If the effects of an amino acid state depend strongly on the states at other sites in the protein – and especially on higher order combinations – then the functions of a protein could become difficult to predict from its sequence or those with similar sequences. Epistasis could also constrain evolution by creating a rugged functional landscape in sequence space, in which many optimal or near-optimal genotypes are inaccessible from others under most evolutionary scenarios (*Breen et al., 2012*; *Carneiro and Hartl, 2010*; *Chen and Conrad, 1994*; *Conrad, 1998*; *Domingo et al., 2019*; *Franke et al., 2011*; *Gavrilets, 1997*; *Gavrilets and Gravner, 1997*; *Goldstein et al., 2015*; *Kauffman and Weinberger, 1989*; *Kauffman and Levin, 1987*; *Kondrashov and Kondrashov, 2015*; *McCandlish et al., 2013*; *Miton et al., 2021*; *Payne and Wagner, 2019*; *Pollock et al., 2012*; *Romero and Arnold, 2009*; *Shah et al., 2015*; *Smith, 1970*; *Starr and Thornton, 2016*; *Usmanova et al., 2015*; *Whitlock et al., 1995*). A related topic is potential bias in epistatic interactions: if they predominantly impair function, 'diminishing the returns' of otherwise favorable mutations (*Chou et al., 2011*; *Cvijović et al., 2018*; *Johnson et al., 2019*; *Kryazhimskiy et al., 2014*; *Lyons et al., 2020*; *Reddy and Desai, 2021*; *Tokuriki et al., 2012*; *Wei and Zhang, 2019*; *Wünsche et al., 2017*) then functional variants might be restricted to only those regions of sequence space where each site has the state or states with the most beneficial main effects, further constraining evolutionary paths.

Epistasis is clearly present in the genetic architecture of proteins (*Ashenberg et al., 2013*; *Bank et al., 2015*; *Bridgham et al., 2009*; *Bridgham et al., 2006*; *Ding et al., 2022*; *Emlaw et al., 2020*; *Faber et al., 2019*; *Gong et al., 2013*; *Gong and Bloom, 2014*; *Kumar et al., 2017*; *Lunzer et al., 2010*; *Ortlund et al., 2007*; *Park et al., 2022*; *Pokusaeva et al., 2019*; *Starr, 1979*; *Starr et al., 2018*; *Tufts et al., 2015*; *Wang et al., 2013*), but its overall character and effects on evolutionary trajectories remain murky. Studies of pairs of mutations show that pairwise interactions can block some two-step paths near a designated wild-type protein (*Adams et al., 2019*; *Diss and Lehner, 2018*; *Olson et al., 2014*; *Salinas and Ranganathan, 2018*) and experiments on higher order combinations have found that epistasis can reduce the number of functional intermediates that connect a designated pair of starting and ending proteins (*Buda et al., 2022*; *de Visser and Krug, 2014*; *Jochumsen et al., 2016*; *Phillips et al., 2021*; *Sailer et al., 2020*; *Sailer and Harms, 2017a*; *Szendro et al., 2013*; *Weinreich et al., 2018*; *Weinreich et al., 2013*). Most combinatorial studies, however, have assessed just two states per site (*Aakre et al., 2015*; *Anderson et al., 2021*; *Anderson et al., 2015*; *Bridgham et al., 2009*; *Bridgham et al., 2006*; *Brown et al., 2010*; *de Visser et al., 2009*; *Dutta et al., 2010*; *Field and Matz, 2010*; *Jenson et al., 2017*; *Jiang et al., 2013*; *Khan et al., 2011*; *Kumar et al., 2017*; *Lee et al., 1997*; *Lozovsky et al., 2009*; *Lunzer et al., 2005*; *Malcolm et al., 1990*; *Meini et al., 2015*; *Miton and Tokuriki, 2016*; *Moriuchi et al., 2014*; *Natarajan et al., 2013*; *Noor et al., 2012*; *O'Maille et al., 2008*; *Palmer et al., 2015*; *Poelwijk et al., 2019*; *Poelwijk et al., 2016*; *Reetz and Sanchis, 2008*; *Tufts et al., 2015*; *Weinreich et al., 2006*; *Yang et al., 2019*; *Zhang et al., 2012*), thus excluding most of the huge ensemble of potential genotypes and the evolutionary paths among them. Further, most of these studies have measured only a single function, even though many protein families contain members with distinct functional specificities, such as the capacity to bind different substrates or other ligands (but see *Moulana et al., 2023*; *Phillips et al., 2023*; *Phillips et al., 2021* for recent exceptions). As a result, the genetic architecture of functional specificity and its influence on the evolution of new functions are poorly understood.

Characterizing the genetic architecture of functional specificity and its evolutionary impact requires three things: (1) an experiment that assays all combinations of 20 amino acid states at a designated set of sites for multiple functions; (2) a method to extract from these data the causal rules by which sequence determines function, and (3) a way to characterize the impact of those rules on trajectories through sequence space. Improved technologies have enabled complete scans of all combinations of 20 amino acids, typically at three or four sites of a priori structural or functional interest (*Jalal et al., 2020*; *Lite et al., 2020*; *McClune et al., 2019*; *Podgornaia and Laub, 2015*; *Raman et al., 2016*; *Starr et al., 2017*; *Wu et al., 2016*; *Yoo et al., 2020*). The key limiting factor has been the lack of effective methods to dissect genetic architecture from such datasets. Most methods for dissecting

genetic architecture from combinatorial studies have been limited to just two states per site (but see *Brookes et al., 2022*; *Faure et al., 2023a*; *Faure et al., 2023b*; *Park et al., 2023*). Another concern is that most studies have modeled the effects of mutations relative to a particular wild-type reference sequence, which yields globally inaccurate estimates of the effects of mutations and combinations when they are introduced into other backgrounds (*Poelwijk et al., 2019*).

Here, we develop and implement a reference-free method for global dissection of 20-state sequence-function relationships and apply it to a combinatorial DMS dataset generated in our laboratory. The model's terms are encoded relative to the global functional average across all genotypes rather than a particular reference sequence, and they directly express the portion of all functional variation in a protein that is attributable to any particular set of genetic determinants. This way of encoding the model also allows us to directly assess the effect of epistasis on the distribution of multiple functions across sequence space and to assess their accessibility under various evolutionary scenarios.

The data we analyze were generated in a complete combinatorial scan at four sites that are critical for DNA recognition in the DNA-binding domain (DBD) of the reconstructed ancestral steroid hormone receptor (*Anderson et al., 2015*; *McKeown et al., 2014*; *Starr et al., 2017*). This experiment assayed the transcription factor's ability to activate transcription from two different biologically relevant DNA response elements, enabling us to characterize the rules of genetic architecture that define functional specificity. By applying our approach to a reconstructed ancestral protein, we were also able to assess how the protein's historical genetic architecture shaped the maintenance of the ancestral function and the accessibility of the derived function that actually evolved during steroid hormone receptor history.

## Results

### Experimental data

Steroid hormone receptors are a family of ligand activated transcription factors that are involved in vertebrate development, behavior, and reproduction (*Bentley, 1998*). There are two major subfamilies of steroid hormone receptors, which recognize distinct palindromic DNA response elements (REs). Estrogen receptors (ERs) bind to a palindrome of the ER response element (ERE, AGGTCA), whereas the ketosteroid receptors (kSRs, including the receptors for androgens, progestogens, glucocorticoids, and mineralocorticoids) strongly prefer the steroid receptor response element (SRE, AGAACA; *Chusacultanachai et al., 1999*; *So et al., 2007*; *Welboren et al., 2009*). Earlier experiments showed that the reconstructed DBD of the last common ancestor of the two subfamilies (called AncSR1) was ERE-specific, whereas the last common ancestor of the kSRs (AncSR2) was SRE-specific (*McKeown et al., 2014*). During the interval between AncSR1 and AncSR2, three substitutions occurred in the protein's recognition helix (RH), which inserts into the DNA major groove and makes contact with the bases that vary between ERE and SRE. These substitutions were shown experimentally to cause the evolutionary change in specificity (*McKeown et al., 2014*).

The data that we analyze here come from a previously published deep mutational scan of AncSR1 (*Starr et al., 2017*). The library contained all 160,000 combinations of 20 amino acids at four sites in the RH – the three historically substituted sites plus another site that is variable in closely related nuclear receptors. The library was transformed into yeast strains carrying either an ERE- or SRE-driven GFP reporter and functionally characterized using a FACS-based Sort-seq assay.

To reduce the effect of measurement noise on the characterization of genetic architecture, we transformed the continuous functional data into categorical form. Each variant was categorized on each RE as a null, weak, or strong activator relative to negative control (stop-codon-containing variants) and historically relevant positive control reference sequences (the RH sequence of AncSR1 and extant ERs on ERE, or that of AncSR2 and extant kSRs on SRE). Variants were classified as null if their mean fluorescence was indistinguishable from the negative controls, weak if they produced fluorescence between null and the historical reference state, and strong if their fluorescence was as great or greater than the reference. Across both REs, 1342 variants were classified as strong and 3166 as weak activators. Categorization substantially reduced the effect of technical noise in the assay: concordance in the activation class assigned to each variant between replicates was >97%, much better than the

between-replicate correlation of mean fluorescence as a continuous variable ($R^2$=0.62 for functional variants, and $R^2$=0.11 when null variants are included).

## Ordinal linear regression of genetic architecture

To dissect the genetic architecture of RE binding and specificity by AncSR-DBD, we developed an ordinal regression model and fit it to the experimental data. The model contains terms for the main effect of every possible amino acid at each variable site in the protein and for the epistatic effect of every pair and triplet of sites (*Figure 1A*). The genetic score of each variant is defined as the sum of the main and epistatic effects of the sequence states and combinations it contains (*Figure 1B*). The variant's genetic score determines the probability that a variant is a null, weak, or strong activator through an ordinal logistic function: an increase in the genetic score causes a corresponding linear increase in the log of the odds that a variant is in a higher vs. lower activation class. The only other free parameters in the model are the average genetic score across all variants and the threshold scores that represent the boundaries between activation classes.

To incorporate functional specificity, the model contains two terms for every amino acid state or combination: one for its effect on nonspecific DNA binding (the contribution to genetic score averaged over ERE and SRE) and another that reflects its impact on specificity (the difference between its score on each RE and the average over REs). The model also contains a single term for the global effect of SRE vs. ERE, averaged across all protein variants, which represents the main effect of the RE itself. The total genetic score for any protein:RE complex is the sum of all the main and epistatic effects of its amino acids on nonspecific DNA binding, plus all the main and epistatic effects on RE specificity, plus the global effect of the RE. (The sign of the specificity and RE terms in the summation is determined by whether the complex contains SRE or ERE.)

The model is encoded to give accurate and efficient estimates of the rules of genetic architecture across the entire set of possible sequences. Rather than defining the effects of a state or combination relative to a designated wild-type sequence, we use a reference-free framework in which effects are defined relative to the global average across all variants (*Figure 1C–D*; for a related approach using DNA sequences, see *Stormo, 2011*). The main effect of any amino acid state is the mean genetic score of all variants containing that state minus the global average score. The epistatic effect of a pair of states is the mean genetic score of all variants containing that pair minus the sum of the global average and the main effects of the two states. And the third-order effect of a triplet of states is the average score of all variants containing that triplet minus the global average and the sum of the pairwise and main effects of the component states and pairs. We did not model fourth-order epistatic effects, because these cannot be distinguished from technical error in the assay measurements.

This approach has several advantages. First, it reduces the effect of estimation bias and measurement error, because each effect is averaged across a large number of observations from unique genotypes at other sites (e.g. 8000 for each main effect term, 400 for each pairwise effect) (*Poelwijk et al., 2019*). Second, the model's form allows us to use an ANOVA-like framework to directly partition the genetic variance – defined as the variance in genetic score across all protein variant-RE complexes – into that attributable to every causal factor in the model. The genetic variance attributable to any sequence state or combination in the model is its squared coefficient, weighted by the fraction of all genotypes to which it applies. The variance explained by a set of terms (e.g. at a site or an epistatic order) is simply the sum of the variance accounted for by all the terms in the set (see Methods and Appendix 1). A third advantage is that the sigmoidal shape of the logistic function may incorporate global nonlinearity in the genotype-phenotype relationship, such as that imposed by limited dynamic range of measurement in the transcriptional assay; this is important, because failing to do so can lead to spurious findings of specific epistatic interactions (*DePristo et al., 2005*; *Domingo et al., 2019*; *Harms and Thornton, 2013*; *Otwinowski et al., 2018*; *Phillips, 2008*; *Sailer and Harms, 2017b*; *Starr and Thornton, 2016*).

We fit the model to the DMS data by least-squares ordinal logistic regression (*Figure 1E*). To reduce overfitting and maximize predictive accuracy, we used L2 regularization with cross-validation to identify the optimal regularization parameters. The model fits the experimental data well. When we used the best-fit model to predict the highest-probability activation class for each protein variant on each RE, the predicted class matched the observed class for 99.5% of all variants. The deviance of the model – an analog of the coefficient of determination used in linear regression – was 0.86

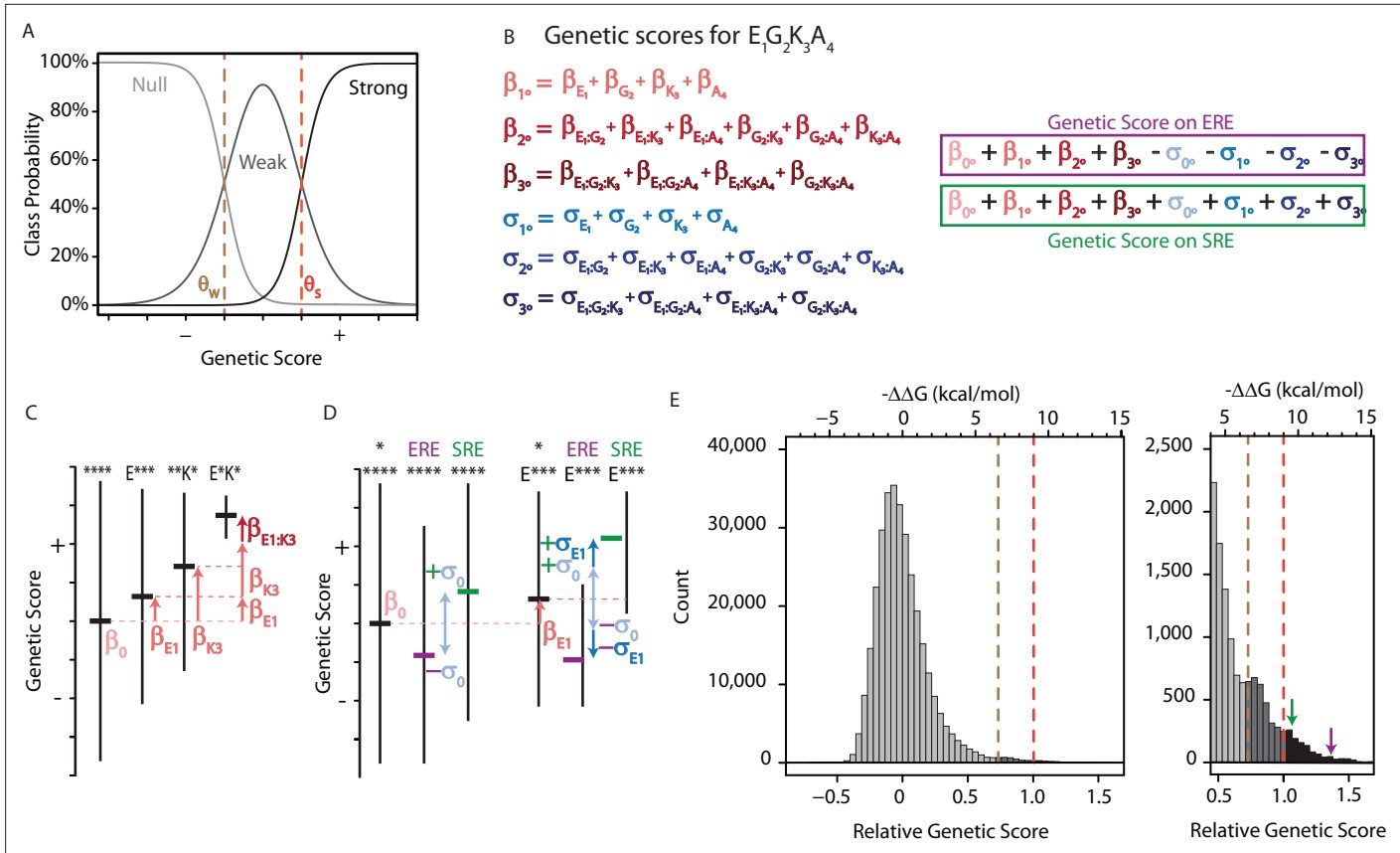

**Figure 1.** An ordinal regression model of the genetic architecture of transcription factor specificity. (**A**) The ordinal logistic model. The probability that a protein variant is in each binding class is a logistic function of its genetic score. Thresholds between classes (dotted lines) correspond to scores at which the probability of being in two classes is equal. (**B**) A variant's genetic score is the sum of the effects of the sequence states and combinations it contains, including main (1°), pairwise (2°), and third-order (3°) effects. Each state and combination has a term for its effect on generic binding averaged over REs (β, red) and another for its effect on specificity (σ, blue) for ERE vs. SRE; specificity terms are added or subtracted to give the genetic score on SRE or ERE, respectively. All terms are defined relative to the global average over all variants (β°). Components of the score for variant EGKA is shown as an example. (**C**) Schematic of reference-free model terms. Each term represents the average deviation of sequences containing a state (or combination) from the sum of the variant's terms at lower orders, including the global average. Vertical and horizontal lines show range and mean of genetic scores for variants containing the specified states; *, any of the 20 amino acid states. (**D**) Reference-free specificity effects. Binding effects (red) are defined the same as in C. The global specificity effect ($\sigma_0$) is the deviation from the global average for all genotypes on ERE (purple) vs. SRE (green). First order specificity effects ($\sigma_1$s) are additional deviations from this global effect between ERE and SRE for genotypes with a particular amino acid at a particular site. (**E**) Distribution of genetic scores after fitting to the experimental DMS data for all 320,000 protein:DNA combinations. Scores were rescaled so that the global average is zero and the threshold score to be classified as a strong binder is 1. Dashed lines, interclass thresholds. Inset, right tail of the distribution. Estimated −ΔΔG values of binding associated with each genetic score are shown along the top, based on calibration to experimental binding data for a subset of variants. Arrows, biological reference sequences GSKV (SRE binder, green) and EGKA (ERE binder, purple).

The online version of this article includes the following figure supplement(s) for figure 1:

**Figure supplement 1.** Model fitting and cross-validation.

**Figure supplement 2.** Model error rates and misclassification.

**Figure supplement 3.** Test of proportional odds assumption.

**Figure supplement 4.** Model fit and ΔG of binding.

**Figure supplement 5.** Model constrained estimates.

(*Figure 1—figure supplement 1*, *Figure 1—figure supplement 2*, *Figure 1—figure supplement 3*). The remaining variation in function not explained by the model is attributable to technical error in measurement and/or fourth-order epistasis. The genetic score of activators is well correlated with the apparent ΔG of dissociation from DNA, which was previously measured for a subset of variants by fluorescence anisotropy ($R^2$=0.75, *Figure 1—figure supplement 4*; this correlation is slightly better than the correlation between the mean fluorescence values used to classify variants and ΔG, $R^2$=0.63).

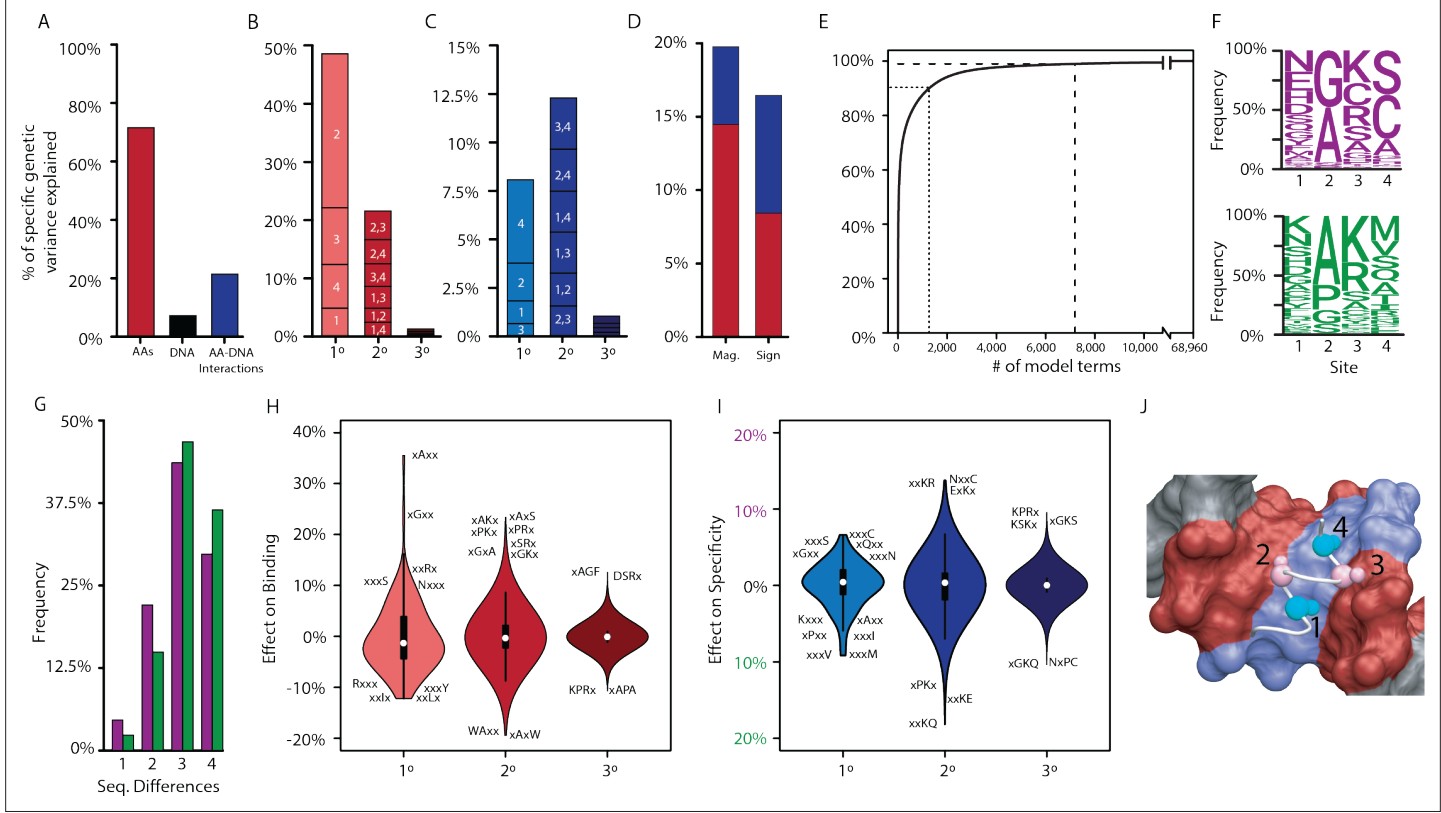

**Figure 2.** Genetic variance explained by components of the model. (**A**) Fraction of genetic variance explained by all terms in the protein's amino acid sequence (non-RE-specific binding, red), the RE's DNA sequence (black), or the interaction between them (specificity, blue). (**B**) Fraction of genetic variance explained by terms in each model order for non-RE-specific binding. Indexes within each column show the variance explained by terms at each site or combination of sites. (**C**) Fraction of genetic variance explained by specificity terms at each order. (**D**) Fraction of genetic variance explained by pairwise epistatic effects that affect only the magnitude or affect the sign of interaction between two states; red and blue, effects on pairwise binding and specificity. (**E**) Cumulative fraction of genetic variance explained by model terms, ordered by the fraction of variance explained by each. Dashed lines indicate 90% and 99% of the total variance in genetic score explained by the complete model. (**F**) Frequency of amino acid states among sequences predicted to be strong activators on ERE (purple) and SRE (green). (**G**) Distribution of amino acid differences among predicted strong activator sequences on ERE (purple) and SRE (green). (**H**) Distributions of effect sizes on non-RE-specific binding. Largest-magnitude effects are labeled by amino acid state (x, any state). (**I**) Distribution of effect sizes on RE specificity. (**J**) Structural view of DBD recognition helix (white tube) and DNA response elements (surface). Blue, nucleotides that differ between ERE and SRE; red, nucleotides identical between ERE and SRE; gray, nucleotides outside of RE. Cα and Cβ atoms of variable residues in the recognition helix are shown as large and small spheres, respectively (cyan, largest effects on RE specificity; pink, largest effects on non-RE-specific binding). Coordinates are from the X-ray crystal structure of AncSR1-DBD, PDB 4OLN.

The online version of this article includes the following figure supplement(s) for figure 2:

**Figure supplement 1.** Number of important coefficients.

**Figure supplement 2.** Variance explained of model coefficients for full model.

**Figure supplement 3.** Correlation among model coefficients.

**Figure supplement 4.** Model coefficients for first and second order.

**Figure supplement 5.** Model coefficients for third order.

## Low-order genetic architecture

We found that main effects and pairwise epistasis are important in RE recognition, but higher order epistasis is not. Of all the genetic variance explained by the best-fit model, more than half is attributable to the main effects of single amino acids, and virtually all of the remainder is attributable to pairwise interactions or the global effect of the RE. Main effects are the primary determinants of nonspecific DNA binding, but pairwise interactions explain the majority of RE-specificity. Third-order epistasis accounts for only 2% of total variance, with similarly small contributions to both nonspecific binding and specificity (*Figure 2A–C*). Of all the specific variance explained by epistatic interactions, about half involves sign epistasis – interactions that on average change the direction of a

state's effects in the presence of another amino acid – while the other half changes the magnitude but not the direction of effect (*Figure 2D*). Although high-order interactions are unimportant, the model is very dense at low orders. Of the nearly 69,000 possible terms in the complete model, it takes >7000 to account for 99% of genetic variance ("the 99% set", *Figure 2E*). This set includes the main effect of every possible amino acid state at all four sites and >80% of all possible second-order epistatic terms. Every amino acid state participates in a minimum of 60 pairwise interactions with states at other sites (*Figure 2—figure supplement 1*). By contrast, the 99% set includes only 5% of all possible third-order terms.

The non-sparsity of the DBD's genetic architecture reflects the fact that thousands of variants activate from ERE and/or SRE, and their sequences are very diverse (*Figure 2F–G*). The effects of most amino acids and pairs are therefore of small to moderate magnitude, changing the genetic score by a median of 3% and 2% of the threshold required for strong activation, respectively. Third-order effects were generally tiny, with a median absolute magnitude of just 0.02% (*Figure 2H–I*, *Figure 2—figure supplement 2*, *Figure 2—figure supplement 3*). As a consequence, no single amino acid state or pair is sufficient to make a sequence into a strong activator; rather, favorable states and combinations of small to moderate size at three or all four of the sites are required to generate a functional protein.

The sequence-function relationship of DNA recognition is therefore complex but not idiosyncratic. The functions of any RH sequence can be predicted with good accuracy from its main and pairwise effects alone (*Figure 2—figure supplement 4*). Higher order epistatic terms are largely dispensable, so prediction does not require precise knowledge of the effects of the vast number of triplets and quartets (*Figure 2—figure supplement 5*). Although low-order prediction is largely sufficient, it

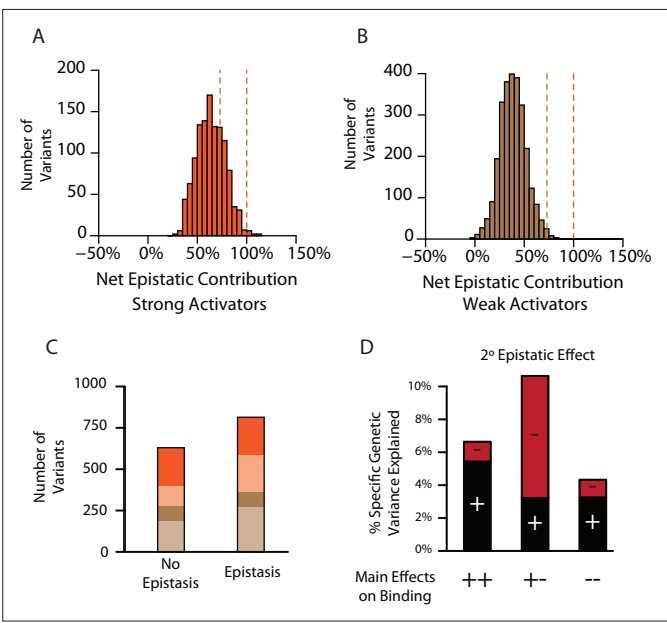

**Figure 3.** Pervasive pairwise epistasis. (**A**) Net contribution of second- and third-order epistatic effects to the total genetic score of all variants predicted to be strong binders, scaled relative to the threshold for classification as a strong binder. Dashed lines, thresholds between activation classes. (**B**) Same as A, but for variants predicted as weak binders. (**C**) Number of variants classified as a weak (brown) or strong (orange) binder using terms from a model fit with no epistasis (1° effects only, left) or the full model with epistasis (1°+2°+3° effects, right). Dark colors represent the number of variants that retain the same class membership in both models; pale colors change class. (**D**) Percent of genetic variance explained by positive (black) and negative (red) pairwise binding interactions, classified by the sign of the main effects of the states.

The online version of this article includes the following figure supplement(s) for figure 3:

**Figure supplement 1.** Net contribution of third-order epistasis to binding.

**Figure supplement 2.** Genotype classifications across models.

**Figure supplement 3.** Variance explained of model coefficients for non-epistatic model.

**Figure supplement 4.** Variance explained by non-specific third order epistasis.

cannot be reduced to a few simple rules, such as 'must have an arginine at site 3' to bind either RE, or 'must have a valine or methionine at site 4' to prefer SRE. Rather, there are hundreds of potential 'and' and 'or' rules by which RH variants across the massive multidimensional space of the protein's sequence can become a specific or nonspecific activator.

The functional effects of variation at each site can to some extent be structurally rationalized (*Figure 2J*). The largest effects of states at sites 2 and 3 are on nonspecific binding, which have their side chains positioned near the parts of the DNA that are shared between ERE and SRE. The most favorable effects come from having small-volume amino acids at site 2, particularly in combination with a basic residue at site 3, which can potentially hydrogen-bond with a guanine common to both REs. By contrast, the largest effects of states at sites 1 and 4 are on specificity, and these residues are closest to the bases in the major groove that differ between the REs. Small hydrophobic residues at site 4 favor SRE binding, where they can pack against the hydrophobic methyl groups that are unique to SRE but leave ERE's hydrogen bond (donors/acceptors) unpaired.

## Epistasis generates functional activators

To determine if epistatic effects have a directional bias on functional sequences, we calculated the net contribution of all epistatic terms to the genetic score of each active variant on each RE (*Figure 3A–B*). Contrary to the expectation of diminishing returns (in which epistasis predominantly impairs function; *Chou et al., 2011*; *Tokuriki et al., 2012*; *Wei and Zhang, 2019*; *Wünsche et al., 2017*), we found that epistatic terms overwhelmingly enhance the function of active variants. Epistatic terms make a net positive contribution to the genetic score of 100% of strong activators, contributing an average of 64% of the score needed to be a strong activator, and the epistatic contribution was necessary to surpass this threshold in every single case. Weak activators also benefit from epistatic interactions, which contribute positively to 98% of variants in this class, but their effect is smaller. Both pairwise and third-order effects contribute, but second-order interactions make a far larger positive contribution than third-order effects do (*Figure 3—figure supplement 1*).

To directly characterize the effect of epistasis on the number, identity, and function of genotypes, we fit to the data truncated first-order models of genetic architecture that allow only main effects and thus exclude all epistasis. We also fit second-order models that allow main and pairwise effects but exclude higher order interactions. We then used each fitted model to predict which variants would be functional activators. Confirming the key role of epistasis in generating functional proteins, we found that the number of strong activators in the main-effects-only model is reduced by about 25% compared to the third-order model, and the number of weak activators is also substantially reduced (*Figure 3C*). Most of this increase is caused by pairwise epistasis, because a second-order model contains 98% as many activators as the third-order model (98%, *Figure 3—figure supplement 2*). Epistasis also changes *which* sequences are functional: of variants that are strong activators in the first-order model, only 64% remain strong activators when epistasis is included.

Why does epistasis increase the number of functional variants and change their identity? Even in the first-order-only model, main effects are mostly of small magnitude (*Figure 3—figure supplement 3*): the only way to achieve a genetic score sufficient for activation is to combine the few amino acids that have the largest main effects, and the number of ways to do this is limited. With epistasis, however, positive epistatic terms can supplement main effects, improving the function of sequences that contain states that have weak main effects. Consistent with this explanation, states with positive main effects on binding overwhelmingly have positive pairwise and third-order epistatic interactions (*Figure 3D*, *Figure 3—figure supplement 4*).

## The genetic architecture of mutation effects

The genetic architecture of a protein directly describes how the sequence of each protein determines its function, but evolution involves mutations – changes that turn one sequence into another. Exchanging one amino acid for another in a particular protein alters the main effect at that site and all the epistatic interactions involving that site. We used our model to characterize the impacts on the genetic score for ERE and SRE binding of all 1.2 million possible context-specific mutations, defined as one of the 1520 mutation types (a single-residue exchange from one of the 20 possible starting residues to 19 ending states at each of four sites) introduced into any of the 8000 possible combinations of states at the other three sites.

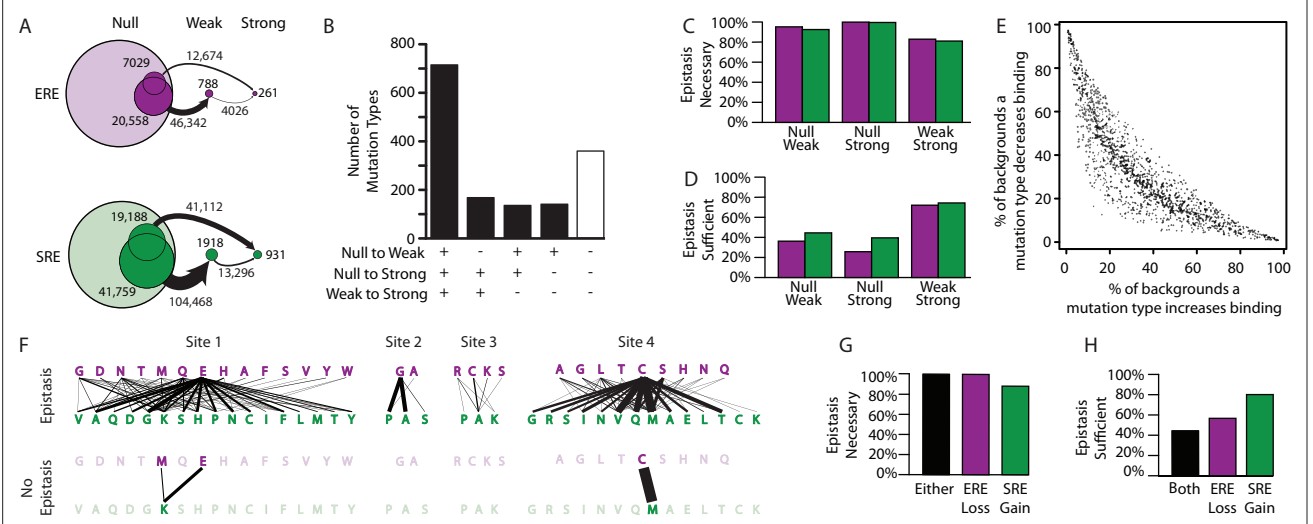

**Figure 4.** Effect of epistasis on the effects of mutations. (**A**) Number of mutations that can promote protein variants from one activation class to another (circles). Each mutation is a single-amino-acid change from one unique sequence into another. (**B**) Effects on activation class of variants by unique mutation types (a change from one amino acid state to another at a particular site, irrespective of the background at other sites). Each column shows the number of mutation types that do (+) or do not (-) promote one or more variants from one class to another. (**C, D**) Of mutations that change a variant's binding class, the fraction is shown for which the contribution of epistasis is necessary (**C**) and/or sufficient (**D**) to cross the threshold for reclassification. (**E**) Context-dependence of mutation effects. Fraction of genetic backgrounds in which a mutation type increases (x-axis) or decreases (y-axis) genetic score. Only mutations that change the genetic score by at least 5% of the threshold to be a strong activator are included. (**F**) Single-amino-acid mutations that change DNA specificity from ERE to SRE in the complete model (top) or the first-order-only model (bottom). Lines show mutations from the state in an ERE-specific variant (purple) to that in the SRE-specific variant (green); thickness is proportional to the number of backgrounds at other sites against which the mutation changes specificity. (**G, H**) Of mutations that change RE specificity, the fraction for which epistasis is necessary (**G**) and/or sufficient (**H**) to lose ERE binding (purple), gain SRE binding (green) or both (black).

The online version of this article includes the following figure supplement(s) for figure 4:

**Figure supplement 1.** Necessity and sufficiency of epistasis.

**Figure supplement 2.** Single step mutations in the pairwise epistasis model.

We found that there are >200,000 context-mutations that can transform a null protein into a functional activator, and epistasis is critical to these transitions (*Figure 4A*). The underlying mutations are diverse, with 1160 of the 1520 unique types of amino acid exchanges moving one or more variants into a higher activation class (*Figure 4B*). In nearly every case (99%), the net epistatic impact of the mutation's effects are positive; in 94% of cases, the epistatic contribution is necessary to move the variant to the higher class, and in almost half of cases it is sufficient (*Figure 4C–D*). Sign epistasis was rampant among mutations: every single one of the 1520 mutation types caused positive effects in some backgrounds and negative effects in others, and more than 90% of mutation types (1380 of 1520) had both positive and negative effects on more than 10% of genetic backgrounds (*Figure 4E*).

Epistasis also expands the mutational opportunities to change the protein's DNA specificity. There are hundreds of context-specific mutations and mutation types that change a strong ERE-specific activator directly into a strong SRE-specific activators. These involve changes at all four sites and use dozens of different amino acid states to generate both ERE and SRE specificity (*Figure 4F*). In every single case, the net change in epistatic effects caused by the mutation is necessary for the switch in specificity, and in almost half of cases the epistatic effects are also sufficient (*Figure 4G–H*). Both pairwise and third-order epistasis contributed: in about half of cases, third-order epistasis is necessary, but it is rarely sufficient (*Figure 4—figure supplement 1*).

To directly test the role of epistasis in creating mutational opportunities to switch DNA specificity, we compared the mutations available under the best-fit third-order model to those under the best-fit first-order model. Without epistasis, the total number of specificity-switching mutations is reduced by >80%, and the number of mutation types declines from 85 to just three (*Figure 4F*). Under a model that incorporates pairwise but not third-order epistasis, the number of RE-switching mutations is intermediate, but closer to that in the third-order model (*Figure 4—figure supplement 2*). Thus, epistasis

– mainly second order – is the primary factor in the opportunity for mutations to change RE-specificity with a single amino acid change.

## Evolutionary paths in sequence space

We next assessed trajectories of functional evolution in sequence space and the effect of epistasis on those trajectories. The RH sequence space consists of all 160,000 possible sequences, each representing one protein variant, with edges connecting nodes that can be directly interconverted by a single nucleotide change given the standard genetic code. We assigned to each node the function(s) predicted by its genetic score under each model. Nonfunctional nodes – those that are null or weak on both REs – were excluded from the network, because purifying selection will remove these genotypes from an evolving population under most circumstances (*Smith, 1969*). We focused our analysis on evolutionary paths from ERE-specific starting points to SRE-specific endpoints, because this is the functional transition that occurred during history.

Sign epistasis is generally thought to constrain epistasis by creating a fragmented sequence space in which local optima are separated by impassable valleys. Sign epistasis is rampant in our datasets (*Figure 2D*, *Figure 4E*), we found that given the best-fit third-order model, 99.8% of functional sequences are connected to each other in a single mutually accessible network (*Figure 5A*).

Epistasis also enlarges the network of connected functional sequences, consistent with the function-enhancing effect of epistatic terms on functional variants. The third-order model contains 1603 functional nodes (1600 of which are connected in a single network), whereas the first-order epistasis-free model contains only 1080 functional nodes. The second-order model is almost as large as the complete third-order model, indicating that pairwise interactions account for most of the network-enlarging effect of epistasis (*Figure 5—figure supplement 1*). In multidimensional sequence space, epistasis therefore does not isolate functional variants: rather, it expands the network of functional sequences, virtually any of which can reach any other through a series of single-nucleotide substitutions.

The probability of reaching any variant from a given starting point depends on the number of substitutions required to reach it. We characterized the evolutionary accessibility of SRE specificity from ERE-specific starting points by simulating evolution from each ERE-specific genotype and measuring the length of the paths required for SRE specificity to evolve. We used two evolutionary scenarios for these simulations. The first represents purifying selection and neutral drift: from any ERE-specific node, every step to any functional neighbor has equal probability, irrespective of which REs it activates, and the path ends as soon as an SRE-specific node is reached (*Figure 5B*). In this scenario, including epistasis shortens the average path length to SRE-specificity for 80% of ERE-specific nodes, with an average reduction of 25% (*Figure 5C*). Accessibility under the second-order model was very similar to the complete model, indicating that the increased accessibility of SRE-specific nodes was primarily caused by pairwise epistasis (*Figure 5—figure supplement 2*).

The second evolutionary scenario incorporates positive selection for SRE-specificity. In this case, the probability of visiting each neighbor is determined by a selection coefficient that is proportional to the node's preference for SRE over ERE. Eliminating epistasis in the first-order model again lengthened the average path taken to SRE-specificity compared to the complete third-order model, although this difference shrank as selection became stronger (*Figure 5D*). This difference again was attributable primarily to pairwise epistasis (*Figure 5—figure supplement 3*).

Epistasis therefore shortens rather than lengthens the paths taken during the evolution of a new function. To understand why this is so, we examined the distribution of functions across sequence space and its effect on the paths that connect ERE- and SRE-specific sequences. We found several contributing factors. First, the minimum distance from ERE-specific starting points to the nearest SRE-specific variant is on average 10% shorter in the third-order model than when epistasis is eliminated in the first-order model (*Figure 5E*). A second factor is that there are far more direct single-step mutations from ERE- to SRE-specific nodes: the average number of SRE-specific neighbors per ERE-specific node is 2.5 times higher in the third-order model, and each SRE-specific variant is also more likely to have at least one ERE-specific single-step neighbor (*Figure 5F*). Third, epistasis reduces the average number of ERE-specific neighbors around ERE-specific nodes, further increasing the probability that a step to SRE specificity will be taken if one is available (*Figure 5F*). Finally, epistasis increases the number of paths by which SRE specificity can evolve: in the third-order model, each ERE-specific node can reach its closest SRE-specific nodes by an average of 65 different paths; excluding epistasis shrinks

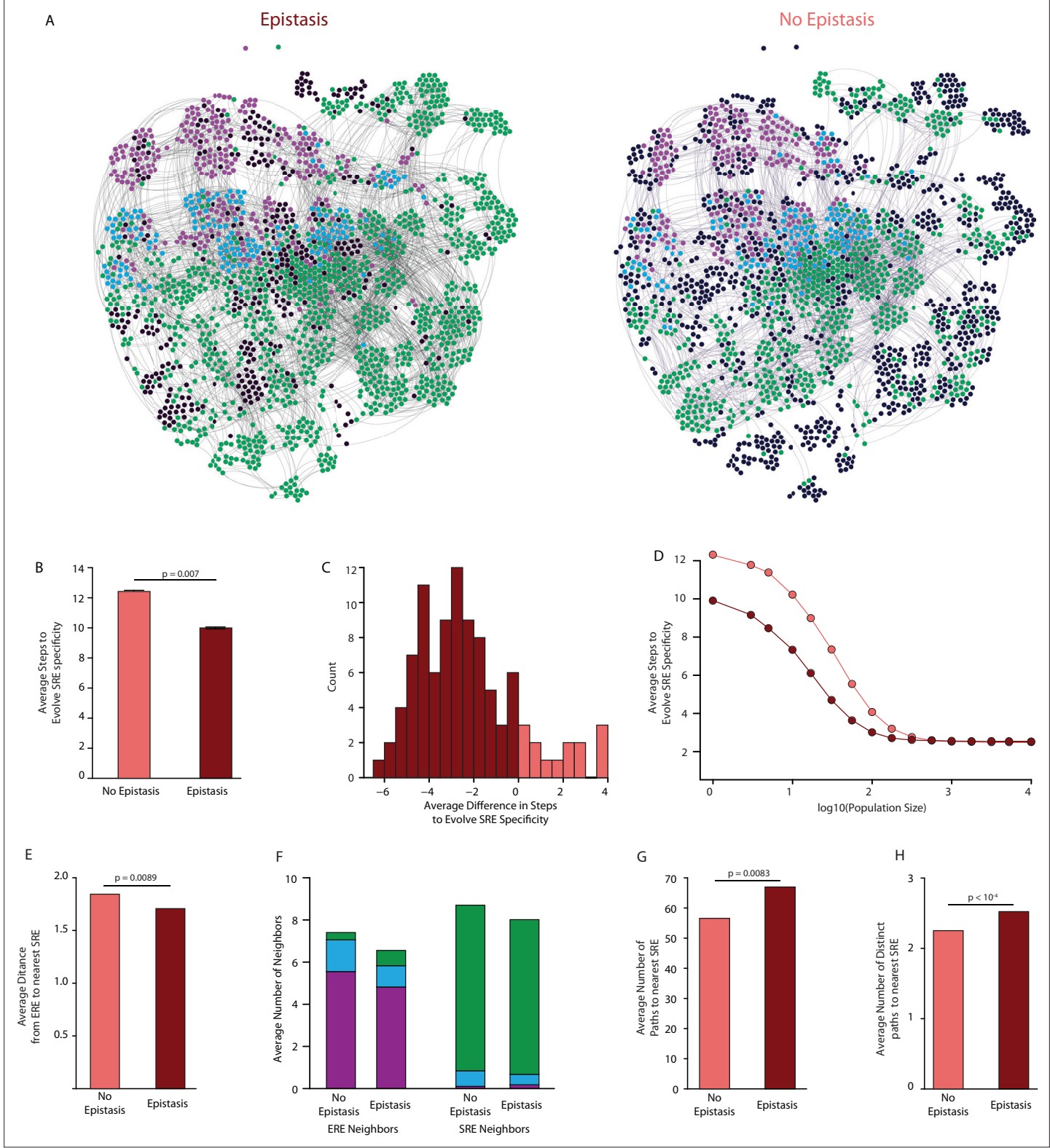

**Figure 5.** Effect of epistasis on the distribution of functions in sequence space. (**A**) Force-directed graph of all variants that are predicted to be strong activators on one or both REs in the complete model (left) or the first-order-only model (right) after fitting to the DMS data. Lines connect variants that are separated by a single nucleotide change given the standard genetic code. Nodes are colored by their predicted functions given each model: strong activator on ERE (purple), SRE (green), or both (cyan); black nodes are null but predicted to be a strong activator under the other model. Nodes that are null in both models are excluded from the network. (**B**) Average number of steps in sequence space for each ERE-specific genotype to reach

*Figure 5 continued on next page*

*Figure 5 continued*

an SRE-specific genotype when evolution is simulated using a model with only purifying selection and drift. Error bars, 95% confidence interval based on 100 replicates per starting point. (**C**) Distribution of the average difference in steps needed to reach an SRE-specific variant from each ERE-specific variant when epistasis is included or excluded, using the same evolutionary process as in B. Light and dark bars show cases in which epistasis makes the path shorter or longer, respectively, from a given ERE-specific variant. (**D**) Average number of steps needed to reach an SRE-specific genotype from each ERE-specific genotype in the sequence space of the complete (dark line) and first-order model (light line), when evolution is simulated with positive selection for increased SRE specificity. The "population size" parameter controls the strength of positive selection relative to drift. (**E**) Average distance from each ERE-specific variant to the closest SRE-specific genotype in the sequence space of the first-order and complete models. (**F**) Average number of single-step neighbors in sequence space per node that is ERE-specific (left) or SRE-specific (right), in the presence or absence of epistasis. Each column shows the average number of neighbors adjacent to the designated node that are strong activators on ERE (purple), SRE (green), or both (cyan). (**G**) Average number of unique minimum length paths connecting ERE-specific variants to their nearest SRE-specific neighbor in the first-order and complete models. (**H**) Same as G for distinct minimum length paths. All p-values were calculated from a permutation test.

The online version of this article includes the following figure supplement(s) for figure 5:

**Figure supplement 1.** Distribution of genotypes and states in sequence space.

**Figure supplement 2.** Evolutionary distances with purifying selection.

**Figure supplement 3.** Evolutionary distances with positive selection for specificity.

**Figure supplement 4.** Effect of epistasis on distribution of functions in sequence space for all models.

this number by about 20%, and these paths are less distinct (*Figure 5G–H*, *Figure 5—figure supplement 4*). Thus, ERE-specific nodes can more easily access nearby SRE-specific nodes when epistasis is present, even though the the *average* distance from ERE specific genotypes to all SRE-specific genotypes increases (*Figure 5—figure supplement 4*), because epistasis adds many new functional variants to the network, and these contain more diverse combinations of states than in the absence of epistasis.

Epistasis therefore facilitates the evolution of new protein functions, rather than constraining it. Epistasis does constrain evolution on a fine scale by making some of the paths between designated pairs of sequences inaccessible. But epistasis also shapes which pairs exist in the network of functional sequences in the first place and how many steps apart those sequences are. When the genetic architecture includes epistasis, there are more functional sequences; also, similar sequences are more likely to have different functions, because a single amino acid difference can combine with the particular residues at the other sites, sometimes dramatically improving one function and impairing the other. By contrast, if the genetic architecture consisted entirely of main effects, variants that share functions would tend to have sequences that are more similar to each other and more different from those with different functions: in this case, each function would be distributed more smoothly across sequence space, and neighboring sequences would be less likely to have distinct functions than when epistasis is present.

## Genetic architecture of a historical change in function

Finally, we sought to understand how the DBD's genetic architecture shaped the historical transition from ERE- to SRE-specificity that occurred during the phylogenetic interval between AncSR1 and AncSR2. Each of the three amino acid changes from the ancestral RH sequence egKa to the derived GSKV (ancestral in lower case, derived in upper case) can be conferred by a single nucleotide change. The functions of the intermediates along the direct path from egKa to GSKV suggest the order in which the substitutions are likely to have occurred: of the three single-mutant variants, only one (GgKa) is functional on either RE – indicating that this was probably the first step – and the same is true of the possible second steps (GgKV being the only accessible functional node, *Figure 6A*).

The ancestral and derived RH variants that occurred during history – and persist to the present – were accessible only because of pairwise epistasis. Under the main-effects only model, neither egKa nor GSKV is a strong activator on either RE. Including the terms from the pairwise or third-order models restores the function of these variants, because interactions make large and necessary contributions to their functions. For example, the glutamate at site 1 in the ancestral variant has virtually no main effect on ERE activation, but it makes a major contribution via pairwise interactions with sites 2 and 3 (*Figure 6B*). Similarly, the serine at site 2 in the derived variant has no main effect on SRE specificity, but its interactions with sites 3 and 4 are necessary to make this protein an SRE activator.

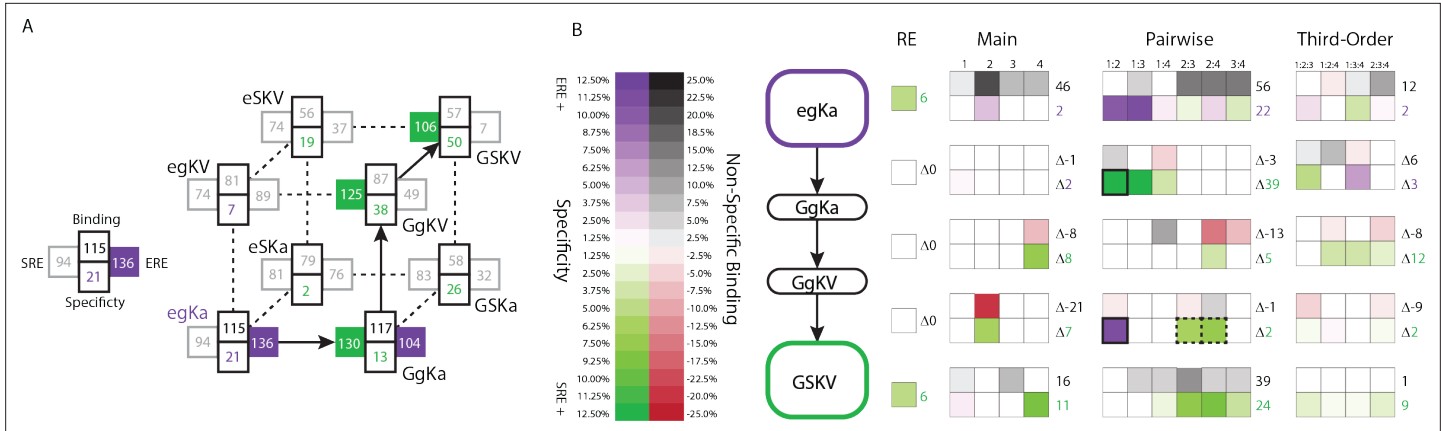

**Figure 6.** Genetic architecture of the historical change in RE specificity. (**A**) Each intermediate on the set of direct paths from the RH sequence of AncSR1 (egKa) to AncSR2 (GSKV) is shown at a vertex of the cube. At each vertex, the total genetic score for binding ERE (right) and SRE (left) are shown, scaled to 100 as the threshold for strong activation, and colored dark if the variant is a strong activator on ERE (purple) and/or SRE (green). Contribution of terms to non-specific binding (top) and specificity (bottom, colored by the RE they favor) are shown. Lines connect genotypes that differ by a single amino acid mutation; solid lines connect vertices that are strong activators on one or both REs; dotted lines connect to a vertex that activates on neither RE. (**B**) Contribution of historical changes in state to the change in RE specificity. Each genotype along the plausible path from AncSR1 to AncSR2 RH is shown at left. For egKa and GSKV, the contribution of effects at each order to the total genetic score for non-RE-specific binding (top) and RE-specificity (bottom) is shown, scaled as in A. Each cell is shaded by the magnitude of the effect at that site or combinations, and the sum of effects for the category is shown at right. The middle rows show the effect of each amino acid mutation along the path, defined as the effect of the derived state (or combination) minus that of the ancestral state or combination. Solid outline, mutation that caused the initial acquisition of SRE binding; dashed outlines, mutations that changed the determinants of SRE binding. The RE column shows the global specificity effect. This value is constant during evolution, but is needed to reconstruct the specificity values shown in A.

These RH sequences are conserved today in all known ERs and kSRs: therefore, the functionality of both historical and present-day steroid receptors depends critically on pairwise epistasis.

The historical transition from ERE- to SRE-specificity also altered the architecture by which REs are recognized, changing the sites and pairs that confer DNA binding rather than just exchanging amino acid states within an otherwise fixed architecture (*Figure 6B*). Moreover, this architecture continued to change once the RH became SRE-specific: the final substitution in the trajectory (g2S, which had no net effect on specificity) replaced the initial historical determinants of SRE specificity with a different set of favorable interactions. As a result, the final RH sequence – which is now found in all extant kSRs – has determinants of function that are entirely different from those in the ancestral RH and even those that first mediated the acquisition of SRE specificity during history.

## Discussion

Many of our findings differ from other studies of the genetic architecture of protein function and evolution. We found only a small role for high-order epistasis, with almost all functional variation among genotypes attributable to main and pairwise effects. Furthermore, the genetic architecture is dense at these low epistatic orders, instead of sparse as found in prior work. Finally, epistasis is not simply a constraint on evolution, but instead facilitates the evolution of new functions by altering which sequences are functional in the first place and how similar sequences with different functions are. These differences arise primarily because we take a global instead of a local view of epistasis, and we expand the scope of analysis to incorporate all possible amino acid states for multiple functions.

### A global view of genetic architecture

Most prior studies have attempted to decompose protein genetic architecture by estimating the effects of mutations (and combinations) on a designated reference sequence (*Buda et al., 2022*; *Lyons et al., 2020*; *Otwinowski et al., 2018*; *Sailer and Harms, 2017a*; *Sailer and Harms, 2017b*; *Weinreich et al., 2018*; *Weinreich et al., 2013*). This strategy provides an accurate local picture of mutation effects when they are introduced into the reference or nearby proteins, but accuracy is poor across other backgrounds and is very sensitive to experimental error (*Poelwijk et al., 2019*). A method

called background averaging reduces this problem by averaging estimates of mutation effects across sets of variants containing a state or combination of interest, but it still defines mutation effects relative to a particular reference state, which may make it sensitive to error in the measurement of variants containing the reference state (*Poelwijk et al., 2019*). In contrast, our approach provides a globally optimal dissection of genetic architecture by minimizing the total deviation between predicted and observed functions across the entire ensemble of variants, regardless of the number of states allowed at each site.

The shift from a reference-based account to a global model of genetic architecture reframes questions about the relationship of sequence to function and its impacts on evolution. In reference-based analyses, the wild-type sequence is taken as a given; the goal is to understand the effect of mutations as the protein moves along paths away from that starting point (usually to a defined endpoint). The starting point itself has no genetic determinants at all. A point mutation can change the function of the original protein via only a single main effect. Never considered are the effect of wild-type states on the protein's functions or a mutation's interactions with those states. That perspective may be appropriate if one is concerned only about the immediate sequence neighborhood of a particular 'wild-type' protein, but it is not well-suited to characterizing a protein's genetic architecture of function or its evolution across an entire sequence landscape.

A reference-free model of genetic architecture, in contrast, asks how sequence states per se – not just a change from a particular beginning state to a particular end state – determine function. The function of each variant in the entire ensemble of sequences is caused by all the states it contains, not by the differences that separate it from a wild-type sequence. Epistasis not only affects the available paths through sequence space from the reference to other nodes: it also determines which nodes are in the space in the first place – the entire set of possible starting, ending, and intermediate genotypes. The effect of mutations is also conceived differently: in the global model, a pair of mutations replaces the main effect of the ancestral states with two new derived main effects, and replaces the original epistatic interactions with every site in the protein – including with those sites at which the state does not change. A mutation does not simply replace one brick in a static genetic architecture; it also changes many of the second-order interactions that determined the protein's functions but that are invisible if the starting point is viewed as a functional black-box. As a protein evolves, each substitution can epistatically change the effect on function of the state at many other sites or modify the effect of other future mutations.

## Low-order, dense genetic architecture

Like other researchers, we found pervasive epistasis within a protein (*Buda et al., 2022*; *Otwinowski et al., 2018*; *Sailer and Harms, 2017a*; *Sailer and Harms, 2017b*; *Weinreich et al., 2018*). Unlike most previous studies, we found that higher-order epistasis plays only a tiny role (*Buda et al., 2022*; *Otwinowski et al., 2018*; *Sailer and Harms, 2017a*; *Sailer and Harms, 2017b*; but see *Weinreich et al., 2018*). We did not measure fourth-order epistasis, but it is unlikely that it will account for much global variation: we obtained good predictions of function from main and pairwise effects alone, with third-order effects explaining very little variance. Moreover, for fourth-order terms to have much of a global effect, each one would have to be considerably larger on average than third-order and second-order terms, but we, like others (*Weinreich et al., 2018*), have observed that effects shrink in magnitude as epistatic order increases. Why do our findings concerning higher order epistasis differ from previous studies? A likely cause is that prior work used reference-based methods and/or they did not adequately account for global nonlinearities. Both of these factors are known to cause spurious inference of higher order epistasis (*Otwinowski et al., 2018*; *Poelwijk et al., 2019*; *Sailer and Harms, 2017b*).

Another difference from previous work is that we found a dense genetic architecture at first and second orders, such that main and pairwise effects involving virtually all possible states at all sites make substantial contributions to genetic architecture. Some other studies have found a very sparse architecture, with just a few states or combinations explaining the distribution of functions across variants (*Poelwijk et al., 2019*). One cause of this difference may be that we analyzed all 20 amino acid states rather than just two, so the number of possible terms in our model is far larger than studies that analyze only pairs of states present in two high-functioning proteins. Consistent with this idea, combinatorial DMS studies have found that there are often numerous functional sequences that share

few to no sequence states at the mutated sites, which necessitates a denser genetic architecture to account for the entire set of functional sequences compared to a set of sequences where only two amino acid states are allowed at each site (*Podgornaia and Laub, 2015*; *Starr et al., 2017*; *Wu et al., 2016*). Another factor may be that all the sites we examined are in the protein's 'active site' – in this case, making direct contact with the DNA to which the protein binds.

## Epistasis and evolution

Prior work has viewed epistasis as an evolutionary constraint, but we found that the primary effect of epistasis is to facilitate functional evolution. Virtually all earlier studies on this topic considered only the effect of epistasis on direct paths through a binary sequence space between two functional sequences designated a priori as starting and ending points, and they have focused on optimization of a single function. In that context, the major effect of epistasis is to create idiosyncrasies in the functions of the intermediates between two high-functioning proteins, so epistasis can only constrain evolution along those paths (*Poelwijk et al., 2007*; *Weinreich et al., 2005*). But this narrow perspective never considers the effect of epistasis in making the designated sequences functional in the first place, and it excludes from view the vast number of additional potential starting, ending, and intermediate nodes in a 20-state combinatorial sequence space.

By incorporating a global perspective, our analysis reveals the facilitating effect of epistasis on evolution. The genetic architecture of function generates a huge network of densely connected functional RH sequences that evolution can navigate (*Bendixsen et al., 2019*; *Payne and Wagner, 2014*). Epistatic interactions are critical in making protein variants functional, so it dramatically increases the number of functional nodes that can be visited during evolution. Moreover, by determining the protein's DNA specificity, epistasis brings variants with different specificities closer together in sequence space; this makes regions of sequence space with new functions far easier to access from those with ancestral functions. By increasing the number of functional nodes and shortening the paths between those with different functions, epistasis dramatically increases the probability that evolution will arrive at a new function from anywhere in the network of potential starting points.

Overall, epistasis makes the topography of sequence space idiosyncratic enough that new functions are easily accessible from nodes with different functions, but not so idiosyncratic that functional nodes become inaccessible from each other. Although epistatic interactions do constrain trajectories when we limit our view to arrival at a particular destination from a particular origin, interactions open opportunities to evolve new functions that are often just a step away from the trajectories that were realized during evolution.

## Limitations and future work

We are aware of several limitations in our study. First, we used an ordinal model to analyze the genetic architecture of functional data transformed into categorical form. This approach could reduce precision in estimating the terms of genetic architecture compared to direct analysis of continuous phenotypic data. The advantage of this approach is that it can reduce the impact of noise. In our dataset, many genotypes are at the lower bound of measurement, and much of the variation in measured florescence for these variants is measurement noise; ordinal regression is therefore expected to reduce the impact of noise on estimates of genetic architecture, and it appears to have done so in this case. Another potential issue is nonspecific epistasis, which if not incorporated can result in inflated estimates of specific epistasis. The logistic model assumes that an amino acid state or combination has a consistent effect on the log-odds of changing a variant's functional category, irrespective of the level of function of the background into which it is introduced. It therefore does not explicitly incorporate nonspecific epistasis, such as that imposed by a limited dynamic range of measurement, although the sigmoid shape may fortuitously accommodate this kind of nonspecific context-dependence. Because our categorical analysis of variants' functional class depends only on the functional rank-order of variants and not on their quantitative function per se, the logistic model is expected to be robust to nonspecific epistasis, so long as the underlying relationship between genotype and phenotypic measurement is monotonic.

We studied four sites in one protein, so the generality of our observations is unknown at this point. For example, genetic architecture away from the protein's DNA-binding surface could be sparser than that in the RH. However, complete single-mutant scans of all sites in other proteins (and the few

studies of all double mutants) shows that most mutations at most sites affect function, consistent with a generally dense genetic architecture – although perhaps less extremely so than in the RH sites that we studied here (*Araya et al., 2012*; *Bank et al., 2014*; *Bloom, 2014*; *Chen et al., 2020*; *DeBartolo et al., 2012*; *Diss and Lehner, 2018*; *Firnberg et al., 2014*; *Fowler et al., 2010*; *Fragata et al., 2018*; *Hietpas et al., 2011*; *McLaughlin Jr et al., 2012*; *Olson et al., 2014*; *Roscoe et al., 2013*; *Salinas and Ranganathan, 2018*; *Sarkisyan et al., 2016*; *Starita et al., 2013*; *Starr et al., 2020*; *Thyagarajan and Bloom, 2014*; *Whitehead et al., 2012*). Higher order epistasis might be more important in other parts of the protein than in the active site, but we know of no structural rationale that might support this expectation: the RH sites are all packed closely in space and bind adjacent nucleotides in the response element, a physical architecture that might be expected to enrich for rather than deplete higher-order effects. We know of no structural or functional reason to expect that high-order epistasis will be more important in other proteins. It is possible that third- (and higher-) order interactions could become more important when larger numbers of sites are considered, because the number of third-order terms that affect each variant will increase as the number of variable sites grows (*Weinreich et al., 2018*). High-order epistasis might also be important in determining multidimensional phenotypes, such as allosteric regulation or cooperativity, that involve multiple protein conformations.

A major challenge in broadening our understanding of protein genetic architecture is the immense size of sequence space. Covering all combinations of states using DMS is currently practical for at most five variable sites at a time, so a complete characterization of genetic architecture at all orders is possible for only tiny protein fragments. Our finding that high-order epistasis is relatively unimportant implies that modeling first- and second-order terms should be sufficient to account for most genetic variation in a protein's functions. If generalizable, the focus could shift from complete sampling of all high-order combinations to covering the much smaller set of pairs. It would still be necessary, however, to analyze the effects of pairs across sufficiently diverse backgrounds at other sites to allow effective global averaging of second-order terms. Evaluating strategies to efficiently accomplish this end without collapsing back into the local tunnel-vision of reference-based, binary state analyses is a key goal for the future.

We found that within the DBD's recognition helix, the relationship between genetic score and its DNA-binding function is tractable and reasonably simple, allowing us to predict the functions of a genotype with good confidence from its main and pairwise effects. To what extent is this finding generalizable from proteins to phenotypes at higher biological levels? Our comprehensive analysis of genetic architecture was possible because we reduced the problem of genetic causality to the effects of a few sites in a single protein domain on a simple biochemical function that can be measured with a high-throughput laboratory assay. Caution is therefore required before extrapolating our results to phenotypes involving multiple molecules or loci (*Bakerlee et al., 2022*). Even in our dataset, most functional specificity is attributable to pairwise interactions between amino acids: this represents third- (or potentially fourth-) order intermolecular epistatic interactions between amino acid pairs and variable nucleotides in the RE. Our assay was also carried out in a controlled and constant environment, and our library was combinatorially complete. This design allowed us to dissect genetic causality free of environmental variance, population structure, gene-environment interactions, or gene-environment correlations – all factors that radically complicate the genotype-phenotype relationship in natural populations. Our study implies that the genetic project may be largely tractable at the level of biochemistry, but it does not imply that complex organismal phenotypes or biosocial traits – particularly those affected by many loci, many environmental variables, and complex dependencies within and between those categories – are predictable by reference to genotype alone.

## Methods
### Experimental data
The experimental data analyzed here were first reported in *Starr et al., 2017*. Libraries of all combinations of all 20 amino acid states at four variable sites in the SR DBD recognition helix (RH, 160,000 total protein variants) were prepared in two different reconstructed ancestral backgrounds: AncSR1 and AncSR1+11P, which contains 11 historical permissive substitutions, which nonspecifically increase DNA affinity without strongly affecting specificity (*McKeown et al., 2014*). The libraries were transformed into yeast strains in which a fluorescent reporter is driven by the ancestral estrogen receptor

response element (ERE) or the derived ketosteroid receptor response element (SRE). Activation by each variant on each RE was measured using a FACS-assisted sort-seq assay and then discretized into three ordered activation categories on each RE: Null if their mean fluorescence was not significantly greater than stop-codon-containing variants, Strong if their mean fluorescence was significantly greater than stop-codon variants and not less than 80% of the historical reference for that RE (AncSR1 on ERE, AncSR1 +11*P*+GSKV on SRE, where GSKV is the sequence of extant ketosteroid receptors in the RH), or Weak if they were neither Null nor Strong. Further details of the categorization can be found in *Starr et al., 2017*. For model fitting, we only used data from the combinatorial library created on the AncSR1 +11 P background as it contained the most balanced number of variants across the three classes.

## Model description

We formulated an ordinal logistic regression model to dissect the effects of amino acid states and combinations on the activation class of RH variants on the two REs (*Figure 1*). The model is a forward-cumulative model with a proportional log-odds assumption – a standard approach to modeling data discretized into ordered classes of an otherwise continuous variable (in this case, mean fluorescence). The model imposes a linear relationship between the predictors (the genetic states/combinations) and the log-odds that a variant is in a set of higher classes vs. a set of lower classes; log-odds is defined as the logarithm of the ratio of the probabilities of the higher vs. the lower classifications. For three activation classes, the relationship between the predicted class $Y(g)$ of a genetic variant with sequence $g$ is:

$$log\left(\frac{P\left(Y\left(g\right)=Null\right)}{P\left(Y\left(g\right)=Weak\,or\,Strong\right)}\right)=\theta_{NW}+\beta X$$

$$log\left(\frac{P\left(Y\left(g\right)=Null\,or\,Weak\right)}{P\left(Y\left(g\right)=Strong\right)}\right)=\theta_{WS}+\beta X$$

where $\theta_{NW}$ and $\theta_{WS}$ are thresholds that delimit the boundaries between the Null/Weak and Weak/Strong binding classes, $X$ is a matrix of indicator variables that represent all the possible sequence states and combinations, and $\beta$ (the variables to be estimated) is a set of coefficients that describe the effect of each state or combination on the log-odds of classification. We evaluated the proportional odds assumption by fitting two simple logistic regression models to the observed activation class data, with variants reclassified as null vs. weak-or-strong, and again as null-or-weak vs. strong. Estimated effects across the two thresholds were similar, as required by the proportional odds assumption (*Figure 1—figure supplement 3*). However, this assumption can be relaxed if there is evidence that the effects of amino acid states or combinations is dependent on the binding class.

We used a reference-free encoding of indicator variables and effect coefficients. First, we describe the form of the model that would be used if only a single phenotype were measured; we then expand the description to incorporate two different phenotypes. For the single-phenotype model, the indicator matrix $X$ describes the amino acid states and combinations contained by each of the 160,000 protein variants. Each row in $X$ specifies a protein variant with sequence $q_1\,q_2\,q_3\,q_4$ at the four variable sites, and each column represents a particular state at a site (e.g. $q_1$=A) or a particular combination of states at a combination of sites (e.g. $q_1q_2$=AA, or $q_1q_2q_3$=AAA). Each cell is assigned value 1 or 0 to specify whether that state or combination is present or absent in the variant. Each coefficient represents the effect of having one of these states or combinations on the log-odds of being in an activation class, relative to the average across all variants. The complete model therefore consists of 80 first-order effects (20 amino acid states at each of the four sites), 2400 second-order effects (400 pairs of states at each of the six pairs of sites), and 32,000 third-order effects (8000 triplets of states at each of the four triplets of sites). A single global intercept, $b_0$ describes the average across all variants (and is associated with indicator variable with value 1 in all variants). In principle, the model could contain all fourth-order combinations and effects, but we did not include them, because each fourth-order coefficient would be estimated from the observed activation class of a single variant and so would be indistinguishable from measurement error.

To incorporate the effects of genetic states on two different functions, the indicator matrix is expanded to uniquely identify all 320,000 protein-RE complexes based on the RE it contains and

the amino acid states/combinations in the protein variant. Each amino acid state/combination is now associated with two coefficients – one for its average effect across the two REs ($\beta$, which represents the RE-nonspecific effect on activation) and the other for the difference in effect on ERE vs. SRE ($\sigma$, representing the effect on specificity). Indicator variables for the RE-nonspecific effects are assigned value 1 if an amino acid state/combination is present in a complex and 0 if absent; indicators for specificity effects are assigned value 1 if the complex contains ERE (and the amino acid state/combination is present), or –1 if it contains SRE. The net effect of any amino acid state or combination on ERE activation is therefore its binding coefficient plus its specificity coefficient; its effect on SRE is the binding coefficient minus the specificity coefficient. A global RE coefficient ($\sigma_0$) represents the main effect of having ERE vs SRE in the complex, averaged over all protein variants (with indicator 1 or –1 for complexes containing ERE or SRE, respectively). As in the single-phenotype model, a global intercept coefficient represents the global average across all complexes ($\beta_0$, with indicator 1 in all complexes).

Although the number of total coefficients in the model is large (68,962 possible), each of the 320,000 complexes is described by just 30 non-zero indicator variables: one for the global intercept, one for the main RE effect, and a binding and specificity coefficient for each of the four amino acid states, six pairs, and four triplets that it contains. For a particular variant with sequence $q_1q_2q_3q_4$ the complete model is specified as follows, after coefficients are multiplied by the relevant indicator variables:

$$\log\left(\frac{P(Y(q_1q_2q_3q_4|ERE) = Null)}{P(Y(q_1q_2q_3q_4|ERE) = Weak\ or\ Strong)}\right) = \theta_{NW} + \beta_0 + \sum_i \beta_{q_i} + \sum_{i<j} \beta_{q_iq_j} + \sum_{i<j<k} \beta_{q_iq_jq_k} + \sigma_0 + \sum_i \sigma_{q_i} + \sum_{i<j} \sigma_{q_iq_j} + \sum_{i<j<k} \sigma_{q_iq_jq_k}$$

$$\log\left(\frac{P(Y(q_1q_2q_3q_4|SRE) = Null)}{P(Y(q_1q_2q_3q_4|SRE) = Weak\ or\ Strong)}\right) = \theta_{NW} + \beta_0 + \sum_i \beta_{q_i} + \sum_{i<j} \beta_{q_iq_j} + \sum_{i<j<k} \beta_{q_iq_jq_k} - \sigma_0 - \sum_i \sigma_{q_i} - \sum_{i<j} \sigma_{q_iq_j} - \sum_{i<j<k} \sigma_{q_iq_jq_k}$$

$$\log\left(\frac{P(Y(q_1q_2q_3q_4|ERE) = Null\ or\ Weak)}{P(Y(q_1q_2q_3q_4|ERE) = Strong)}\right) = \theta_{WS} + \beta_0 + \sum_i \beta_{q_i} + \sum_{i<j} \beta_{q_iq_j} + \sum_{i<j<k} \beta_{q_iq_jq_k} + \sigma_0 + \sum_i \sigma_{q_i} + \sum_{i<j} \sigma_{q_iq_j} + \sum_{i<j<k} \sigma_{q_iq_jq_k}$$

$$\log\left(\frac{P(Y(q_1q_2q_3q_4|SRE) = Null\ or\ Weak)}{P(Y(q_1q_2q_3q_4|SRE) = Strong)}\right) = \theta_{WS} + \beta_0 + \sum_i \beta_{q_i} + \sum_{i<j} \beta_{q_iq_j} + \sum_{i<j<k} \beta_{q_iq_jq_k} - \sigma_0 - \sum_i \sigma_{q_i} - \sum_{i<j} \sigma_{q_iq_j} - \sum_{i<j<k} \sigma_{q_iq_jq_k}$$

where | indicates the RE in the complex, and $i$, $j$, and $k$ index sites in the RH sequence.

## Specific vs non-specific epistasis and the genetic score

In general, epistatic interactions among states can be classified as specific or non-specific (*Harms and Thornton, 2013*; *Starr and Thornton, 2016*). Non-specific epistasis occurs if a global non-linear relationship between an underlying physical property and the measured phenotype affects all combinations of states in the same way. Specific epistasis occurs when combinations of states have distinct non-additive effects on the underlying physical properties. Failure to account for non-specific epistasis when modeling epistatic interactions can artificially inflate inferences of specific epistasis (*Otwinowski et al., 2018*; *Sailer and Harms, 2017b*). Nonspecific epistasis can arise from intrinsically nonlinear relationships between physical properties and measured phenotypes (such as the energy of binding and occupancy of the bound state), limited dynamic range in an assay or biological phenotype, nonlinear scaling within the dynamic range, or the imposition of thresholds within the dynamic range for classifying the functionality of variants.

Our model estimates specific and non-specific epistatic effects in a single fitting procedure by including both forms of epistasis in the model. We do so by defining the genetic score $y(g)$ of any variant $g$ with sequence $q_1q_2q_3q_4$ in complex with ERE or SRE as the sum of all the specific genetic effects of the states in $g$ and the complex:

$$y(g|ERE) = \beta_0 + \sum_i \beta_{q_i} + \sum_{i<j} \beta_{q_iq_j} + \sum_{i<j<k} \beta_{q_iq_jq_k} + \sigma_0 + \sum_i \sigma_{q_i} + \sum_{i<j} \sigma_{q_iq_j} + \sum_{i<j<k} \sigma_{q_iq_jq_k}$$

$$y(g|SRE) = \beta_0 + \sum_i \beta_{q_i} + \sum_{i<j} \beta_{q_iq_j} + \sum_{i<j<k} \beta_{q_iq_jq_k} - \left(\sigma_0 + \sum_i \sigma_{q_i} + \sum_{i<j} \sigma_{q_iq_j} + \sum_{i<j<k} \sigma_{q_iq_jq_k}\right)$$

The specific epistatic interactions are estimated as the second- and third-order epistatic terms of the model, which have an additive effect on the genetic score (together with all the non-epistatic

first- and zero-order effects). Nonspecific epistasis is then incorporated through the logit function, which makes classification a nonlinear outcome of the genetic score.

## Epistatic coding and interpreting coefficient estimates

Most efforts to model sequence-function relationships have encoded genetic effects as the effects of mutations (singly or in combination) relative to a designated reference or 'wild-type' sequence. Others have used a Fourier transformation to recode genotypes and estimate the effects of the transformed states relative to the average across all genotypes (*Brookes et al., 2022*; *Poelwijk et al., 2016*; *Stormo, 2011*). Here, we develop a reference-free approach that directly estimates the effect of any amino acid state or combination on the genetic score relative to the global average across all variants. In our model, the zero-order term is the global average:

$$\beta_0 = \frac{1}{2 \cdot 20^4} \sum_{g \epsilon G} y\left(g\right) = \overline{y\left(G\right)}$$

where $y(g)$ is the genetic score of a particular genotype $g$ and $G$ is the set of all possible protein sequence/RE element complexes; the factor before the sum averages over all protein genotypes times 2 to reflect the number of REs. The main effect of a particular amino acid state $q$ at a particular site $i$ on the log-odds of activation (not specific to an RE) is the average difference between the genetic score of all variants with that state and the global average:

$$\beta_{q_i} = \overline{y\left(G_{q_i}\right)} - \beta_0 = \frac{1}{2 \cdot 20^3} \sum_{g \epsilon G_{q_i}} y\left(g\right) - \beta_0$$

where $G_{q_i}$ is the set of genotypes with amino acid state $q$ at site $i$. Similarly, a pairwise epistatic effect is the difference between the average genetic score of $G_{q_i q_j}$ – the set of variants containing states $q_i$ and $q_j$ at a particular pair of sites $i$ and $j$ – and the expected score if there was no pairwise interaction:

$$\beta_{q_i q_j} = \overline{y\left(G_{q_i q_j}\right)} - \left(\beta_0 + \beta_{q_i} + \beta_{q_j}\right) = \frac{1}{2 \cdot 20^2} \sum_{g \epsilon G_{q_i q_j}} y\left(g\right) - \left(\beta_0 + \beta_{q_i} + \beta_{q_j}\right)$$

Third-order interactions are the difference between the expected genetic score based on the relevant lower-order effects:

$$\beta_{q_i q_j q_k} = \overline{y\left(G_{q_i q_j q_k}\right)} - \left(\beta_0 + \beta_{q_i} + \beta_{q_j} + \beta_{q_k} + \beta_{q_i q_j} + \beta_{q_i q_k} + \beta_{q_j q_k}\right)$$
$$= \frac{1}{2 \cdot 20} \sum_{g \epsilon G_{q_i q_j q_k}} y\left(g\right) - \left(\beta_0 + \beta_{q_i} + \beta_{q_j} + \beta_{q_k} + \beta_{q_i q_j} + \beta_{q_i q_k} + \beta_{q_j q_k}\right)$$

The specificity coefficients are encoded similarly. The global RE coefficient reflects the average difference between the genetic score of all complexes containing ERE and the global average (which is of equal magnitude but opposite sign as the average difference of all complexes containing SRE from the global average). This equals half the difference between the average genetic scores for all complexes on ERE and all complexes on SRE:

$$\sigma_0 = \overline{y\left(G^E\right)} - \overline{y\left(G\right)} = -\left(\overline{y\left(G^S\right)} - \overline{y\left(G\right)}\right) = \frac{1}{2}\left(\overline{y\left(G^E\right)} - \overline{y\left(G^S\right)}\right)$$
$$= \frac{1}{20^4}\left(\sum_{g \epsilon G^E} y\left(g^E\right) - \sum_{g \epsilon G^S} y\left(g^S\right)\right)$$

where $y\left(g^E\right)$ is the genetic score when a protein variant with sequence $g$ is bound to ERE and $y(g^S)$ is the genetic score when a genotype $g$ is bound to SRE. The remaining specificity coefficients follow the same pattern, reflecting differences in specificity beyond what is expected from lower-order effects. For the main specificity coefficients:

$$\sigma_{q_i} = \frac{1}{2}\left(\overline{y\left(G_{q_i}^E\right)} - \overline{y\left(G_{q_i}^S\right)}\right) - \sigma_0$$

For pairwise specificity coefficients:

$$\sigma_{q_i q_j} = \frac{1}{2} \left( \overline{y\left(G_{q_i q_j}^E\right)} - \overline{y\left(G_{q_i q_j}^S\right)} \right) - \left( \sigma_0 + \sigma_{q_i} + \sigma_{q_j} \right)$$

For third-order specificity coefficients:

$$\sigma_{q_i q_j q_k} = \frac{1}{2} \left( \overline{y\left(G_{q_i q_j q_k}^E\right)} - \overline{y\left(G_{q_i q_j q_k}^S\right)} \right) - \left( \sigma_0 + \sigma_{q_i} + \sigma_{q_j} + \sigma_{q_k} + \sigma_{q_i q_j} + \sigma_{q_i q_k} + \sigma_{q_j q_k} \right)$$

## Model selection, regularization, and cross-validation

We fit the model to the DMS data using regularized logistic regression in R (*R Development Core Team, 2023*). To avoid overfitting model parameters to measurement noise, we used regularization via ridge regression (*L2* norm) (*Figure 1—figure supplement 1*). To identify the optimal regularization penalty lambda, we used iterated 10-fold cross-validation. Specifically, we fit a series of models with 900 lambda values of decreasing magnitude from $10^{-1}$ to $10^{-8}$ on a $\log_{10}$ scale. The data were randomly divided into 10 equally sized blocks, and the model at each lambda was fit to 90% of the data, leaving out 10% of the data as a test set, and the activation class of each variant in the test set was predicted on each RE. We compared these predictions to the observed class for each variant, counting misclassification into an adjacent category (weak-null or weak-strong) as a single error and misclassification between null and strong as two errors. We repeated this procedure, using each of the 10 blocks as the test set in turn and calculated the mean misclassification rate. We then repeated this entire procedure ten times, dividing the data into different blocks each time, and using the variation in the estimated misclassification ratio to determine the standard error in our estimate. We applied this procedure to all values of lambda and selected the value with the lowest average misclassification rate.

## Model implementation

To implement the cumulative odds model with three ordered classes and the proportional odds assumption in a framework that allowed for regularization and cross-validation, we made several modifications to the glmnetcr function in the glmnetcr package (*Archer, 2010*).

### Proportional odds assumption via logistic regression

In general, the cumulative odds model can be viewed as generating a set of binary variables around each threshold, each of which distinguishes between the set of classes lower and the set of classes higher than the threshold. For our three activation classes, we recoded the categories using two binary variables – one that distinguishes the Null class from Weak-or-Strong and another that distinguishes Null-or-Weak from Strong. Null corresponds to the 00 class, weak to the 10 class, and strong to the 11 class. Logistic regression can then be used to estimate the best-fit model parameters that predict these recoded classes. This produces equivalent estimates of model coefficients as ordinal regression.

### Sparse matrix representation

Manipulating the large model matrix imposed a large computational burden. Since a large portion of the matrix consists of zeros (each row contains only 30 non-zero values), we used the MatrixModels R package (*Bates and Maechler, 2022*) and modification of functions in the glmnetcr package to implement the handling of sparse matrices.

### Long vector accommodation

The version of R we used fit logistic regression models with regularization using Fortran. To accommodate the large number of observations and covariates, we modified the lognet function from the glmnet R package to handle long vectors and passed this to the underlying Fortan code using the dotCall64 R package (*Gerber et al., 2018*).

## Zero-sum constraint on coefficients

The reference-free framework requires the sum of model coefficients for a given site (or combination of sites) to sum to zero. All marginal subsets of pairwise or third-order coefficients – defined as the subset of coefficients for a given pair (or triplet) of sites that share a particular state (or pair of states) – must also sum to zero. For example, of the 400 pairwise interactions between sites 1 and 2, the entire

set must sum to zero; of these, 20 have an A at site 1, and this marginal set must also sum to zero. This constraint guarantees that all coefficients represent the average effect of having some sequence state or combination relative to prediction from applicable lower-order terms and the global average.

Imposing this constraint during regularized regression is computationally expensive. We therefore performed unconstrained regularized regression and then adjusted the unconstrained coefficients post-hoc to impose this constraint. Specifically, we calculated the average of each set of unconstrained coefficients and each marginal set of unconstrained coefficients. We then subtracted the relevant averages from each unconstrained coefficient, thus recentering each set and marginal subset around an average of zero. For third-order interactions, the constrained estimates are:

$$\hat{\beta}_{q_i q_j q_k} = \beta_{q_i q_j q_k} - \left( \overline{\beta_{*_i q_j q_k}} + \overline{\beta_{q_i *_j q_k}} + \overline{\beta_{q_i q_j *_k}} \right) + \left( \overline{\beta_{*_i *_j q_k}} + \overline{\beta_{*_i q_j *_k}} + \overline{\beta_{q_i *_j *_k}} \right) - \overline{\beta_{*_i *_j *_k}}$$

$$\hat{\sigma}_{q_i q_j q_k} = \sigma_{q_i q_j q_k} - \left( \overline{\sigma_{*_i q_j q_k}} + \overline{\sigma_{q_i *_j q_k}} + \overline{\sigma_{q_i q_j *_k}} \right) + \left( \overline{\sigma_{*_i *_j q_k}} + \overline{\sigma_{*_i q_j *_k}} + \overline{\sigma_{q_i *_j *_k}} \right) - \overline{\sigma_{*_i *_j *_k}}$$

where $\hat{\beta}$ and $\hat{\sigma}$ are the constrained coefficients, $\beta$ and $\sigma$ the unconstrained coefficients, and the averages are over the possible states at a site (or combination) indicated by the $*$. Pairwise coefficients follow the same pattern, with the total marginal effect of each combination of amino acids to be zero:

$$\hat{\beta}_{q_i q_j} = \beta_{q_i q_j} - \left( \overline{\beta_{*_i q_j}} + \overline{\beta_{q_i *_j}} - \overline{\beta_{*_i *_j}} \right) + \left( \overline{\beta_{q_i q_j *_k}} - \overline{\beta_{q_i *_j *_k}} - \overline{\beta_{*_i q_j *_k}} + \overline{\beta_{*_i *_j *_k}} \right)$$
$$+ \left( \overline{\beta_{q_i q_j *_l}} - \overline{\beta_{q_i *_j *_l}} - \overline{\beta_{*_i q_j *_l}} - \overline{\beta_{*_i *_j *_l}} \right)$$

$$\hat{\sigma}_{q_i q_j} = \sigma_{q_i q_j} - \left( \overline{\sigma_{*_i q_j}} + \overline{\sigma_{q_i *_j}} - \overline{\sigma_{*_i *_j}} \right) + \left( \overline{\sigma_{q_i q_j *_k}} - \overline{\sigma_{q_i *_j *_k}} - \overline{\sigma_{*_i q_j *_k}} + \overline{\sigma_{*_i *_j *_k}} \right)$$
$$+ \left( \overline{\sigma_{q_i q_j *_l}} - \overline{\sigma_{q_i *_j *_l}} - \overline{\sigma_{*_i q_j *_l}} - \overline{\sigma_{*_i *_j *_l}} \right)$$

Similarly for the main effect terms:

$$\hat{\beta}_{q_i} = \beta_{q_i} - \overline{\beta_{*_i}} + \left( \overline{\beta_{q_i *_j}} - \overline{\beta_{*_i *_j}} \right) + \left( \overline{\beta_{q_i *_k}} - \overline{\beta_{*_i *_k}} \right) + \left( \overline{\beta_{q_i *_l}} - \overline{\beta_{*_i *_l}} \right)$$
$$+ \left( \overline{\beta_{q_i *_j *_k}} - \overline{\beta_{*_i *_j *_k}} \right) + \left( \overline{\beta_{q_i *_j *_l}} - \overline{\beta_{*_i *_j *_l}} \right) + \left( \overline{\beta_{q_i *_k *_l}} - \overline{\beta_{*_i *_k *_l}} \right)$$

$$\hat{\sigma}_{q_i} = \sigma_{q_i} - \overline{\sigma_{*_i}} + \left( \overline{\sigma_{q_i *_j}} - \overline{\sigma_{*_i *_j}} \right) + \left( \overline{\sigma_{q_i *_k}} - \overline{\sigma_{*_i *_k}} \right) + \left( \overline{\sigma_{q_i *_l}} - \overline{\sigma_{*_i *_l}} \right)$$
$$+ \left( \overline{\sigma_{q_i *_j *_k}} - \overline{\sigma_{*_i *_j *_k}} \right) + \left( \overline{\sigma_{q_i *_j *_l}} - \overline{\sigma_{*_i *_j *_l}} \right) + \left( \overline{\sigma_{q_i *_k *_l}} - \overline{\sigma_{*_i *_k *_l}} \right)$$

The intercepts then capture the average deviations from zero:

$$\hat{\beta}_0 = \beta_0 + \overline{\beta_{*_1}} + \overline{\beta_{*_2}} + \overline{\beta_{*_3}} + \overline{\beta_{*_4}} + \overline{\beta_{*_1 *_2}} + \overline{\beta_{*_1 *_3}} + \overline{\beta_{*_1 *_4}} + \overline{\beta_{*_2 *_3}} + \overline{\beta_{*_2 *_4}} + \overline{\beta_{*_3 *_4}} + \overline{\beta_{*_1 *_2 *_3}} + \overline{\beta_{*_1 *_2 *_4}} +$$
$$\overline{\beta_{*_1 *_3 *_4}} + \overline{\beta_{*_2 *_3 *_4}}$$

$$\hat{\sigma}_0 = \sigma_0 + \overline{\sigma_{*_1}} + \overline{\sigma_{*_2}} + \overline{\sigma_{*_3}} + \overline{\sigma_{*_4}} + \overline{\sigma_{*_1 *_2}} + \overline{\sigma_{*_1 *_3}} + \overline{\sigma_{*_1 *_4}} + \overline{\sigma_{*_2 *_3}} + \overline{\sigma_{*_2 *_4}} + \overline{\sigma_{*_3 *_4}} + \overline{\sigma_{*_1 *_2 *_3}} + \overline{\sigma_{*_1 *_2 *_4}} +$$
$$\overline{\sigma_{*_1 *_3 *_4}} + \overline{\sigma_{*_2 *_3 *_4}}$$

This procedure does not change the genetic score of any variant, and it does not change the likelihood of any model. Constraining the model this way merely centers and scales coefficients so that each represents the average effect of an amino acid or interaction relative to the average of the entire data set (and to the average predicted by lower order terms). In practice, we found that L2 regularization brings the average of most sets of coefficients close to zero anyway, so the post-hoc constraint changes most terms by less than 1% of the unconstrained estimates (*Figure 1—figure supplement 5*).

## Alternative models without epistasis

To understand the effects of specific epistasis on protein function and evolution, we compared the third-order model described above to models that do not incorporate any epistasis, and thus contain only first-order terms (first-order model), or no higher order epistasis, and thus contains only first- and second-order terms (second-order model). To implement these lower order models, we modified the matrix of indicator variables by removing all indicators for combinations at the orders to be excluded. These models were then fit to the data using analogous selection procedures for the full model containing first-, second-, and third-order terms.

## Comparison to observed DMS data

To estimate the quality of the model fit, we compared the observed class of each protein variant to the observed classification from the empirical data. In each case, we calculated the number of true positives, true negatives, false positives, and false positives for each of the three activation categories, as well as a 'non-strong' category (union of null and weak) and an 'activators' category (union of weak and strong) (*Figure 1—figure supplement 2*).

## Comparison to measured ΔG

We compared the estimated genetic score for a subset of protein variants to their energy of binding to ERE and SRE, previously measured using fluorescence anisotropy (*Anderson et al., 2015*; *Figure 1—figure supplement 4*). We used simple least-squares linear regression to estimate the relationship between the genetic score and the ΔG of binding. Because both genetic score and ΔG accrue additively, the relationship between these quantities should be linear. The slope and intercept of the relationship between genetic score and ΔG was then used it to estimate the ΔG of binding for all variants in the data set from their genetic score.

## Model variance explained by individual terms

The reference-free coding of model coefficients leads to a simple relationship between the effect size of a term and the fraction of genetic variance in phenotype that it explains. A detailed proof is provided in Appendix 1 and summarized here.

The genetic variance is defined as the variance in phenotype attributable to all the effects of individual genetic states and combinations; it excludes genetic variance caused by nonspecific epistasis (which is captured by the logit function in our model). The total genetic variance is the average squared deviation of each variant's genetic score from the global mean:

$$Var\left(y\left(G\right)\right) = \frac{1}{2 \cdot 20^4} \sum_{g \in G} \left(y\left(g\right) - \overline{y\left(G\right)}\right)^2$$

where $G$ is the set of all protein variants on either RE, $y(g)$ is the genetic score of a particular protein-RE complex, and $2 \times 20^4$ is the total number of complexes.

The total variance explained by a model can be partitioned into the partial variances explained by each causal factor. In the case of our model, the total genetic variance can be partitioned into the partial variances attributable to the possible states at each site or (combination of states at multiple sites). Organizing these partial variances by the epistatic order and effect type (nonspecific or RE-specific):

$$Var\left(y\left(G\right)\right) = \sum_i Var\left(\beta_i\right) + \sum_{i<j} Var\left(\beta_{i,j}\right) + \sum_{i<j<k} Var\left(\beta_{i,j,k}\right) + Var\left(\sigma_0\right) + \sum_i Var\left(\sigma_i\right) + \sum_{i<j} Var\left(\sigma_{i,j}\right) + \sum_{i<j<k} Var\left(\sigma_{i,j,k}\right) \quad (1)$$

where $i$, $j$, and $k$ index the four variable sites, the variances are over all the possible states (or combinations), and the summations are over all sites (or combination).

The model coefficients have a straightforward relationship to this partitioned variance. Main-effect coefficients are defined as the average deviation of a subset of variants containing a state or combination relative to the global mean, and higher-order coefficients are defined as deviations from the sum of lower-order terms. The variance attributable to any state (or combination) is therefore simply the square of its coefficients, and the variance attributable to any set of states is the average of the squared coefficients (see proof in Appendix 1). Thus, for any given site $i$ (or site combination $i,j$ or $i,j,k$), the variance attributable to the set of model terms applicable to that site can be written as follows:

$$Var\left(\beta_i\right) = \frac{1}{20}\sum_q \beta_{q_i}^2$$

$$Var\left(\beta_{i,j}\right) = \frac{1}{20^2}\sum_{q^2} \beta_{q_iq_j}^2$$

$$Var\left(\beta_{i,j,k}\right) = \frac{1}{20^3}\sum_{q^3} \beta_{q_iq_jq_k}^2$$

$$Var\left(\sigma_0\right) = \sigma_0^2 \qquad (2)$$

$$Var\left(\sigma_i\right) = \frac{1}{20}\sum_q \sigma_{q_i}^2$$

$$Var\left(\sigma_{i,j}\right) = \frac{1}{20^2}\sum_{q^2} \sigma_{q_iq_j}^2$$

$$Var\left(\sigma_{i,j,k}\right) = \frac{1}{20^3}\sum_{q^3} \sigma_{q_iq_jq_k}^2$$

The leading factors in *Equation 2* can all be expressed as a function of *O,* the epistatic order of the effect they represent, where *O*=0 for the global average effect of the RE, and 1, 2, or 3 for the main, pairwise, and third-order epistatic effects of each amino acid combination. Substituting into *Equation 1*, the total genetic variance is therefore the sum of the squared effects of every model term at every order (except for the global average), each divided by the number of model terms at that order:

$$Var\left(y\left(G\right)\right) = \sum_{\beta,O\neq 0} \frac{\beta^2}{20^{O(\beta)}} + \sum_\sigma \frac{\sigma^2}{20^{O(\sigma)}}$$

We can use this same approach to calculate $F\left(Var\left(\theta\right)\right)$, the fraction of the total genetic variance explained by any coefficient $\theta$ (a particular $\beta$ or $\sigma$ of any order, except the global average):

$$F\left(Var\left(\theta\right)\right) = \frac{\dfrac{\theta^2}{20^{O(\theta)}}}{Var\left(y\left(G\right)\right)}$$

The fraction of genetic variance explained by any set of coefficients is simply the sum of $F\left(Var\left(\theta\right)\right)$ over all the coefficients in the set.

This formalizes the intuition that within an epistatic order, model terms of large magnitude explain more variation than smaller terms; if two terms at different order have the same magnitude, the lower order term explains more variation than the higher-order term, because the former affects more genotypes than the latter. The leading terms effectively weight each set of coefficients in a set by the number of genotypes to which they apply.

We used this simple relationship between the magnitude of a model coefficient, its order of epistatic effect, and the fraction of genetic variance explained by that coefficient to directly calculate the percent of variance explained by every model term. We then summed the percent variance explained by individual terms to determine the percent of variance explained by sets of terms (*Figure 2*). Finally, we used this relationship to define the 'important' terms in the model as those terms needed to explain 99% of the model variance. Thus setting all of the 'unimportant' terms to zero reduces the amount of variance explained by less than 1% of the full model (*Figure 2—figure supplement 1*).

## Fraction of genetic score attributable to epistasis or specificity

To find the relative impact of epistasis terms on each variant's genetic score, we calculated the sum of the absolute value of all pairwise and third-order model terms for that genotype and divided it by the sum of the absolute value of all model terms for that genotype (*Figure 3*; *Figure 3—figure supplement 1*; *Figure 3—figure supplement 2*). For specificity, we summed the absolute value of only the specificity terms divided by the total absolute sum of all terms.

## Types of epistasis

Epistasis can be divided into several subtypes depending on the signs and magnitude of mutational effects on different genetic backgrounds. These distinctions are most commonly made for pairwise epistatic interactions. For example, sign epistasis arises when the direction of effect of a mutation

depends on the genetic background. Reciprocal sign epistasis occurs when the direction of effect of both mutations depends on the genetic background. All other instances of epistasis are classified as magnitude epistasis. Within magnitude epistasis, there are two broad forms, diminishing returns and diminishing costs epistasis. The former, diminishing returns, also includes common forms of epistasis such as amplifying costs, differing only on which genotype is considered 'wild-type' or the starting point of a mutant cycle. Likewise, diminishing costs epistasis also covers amplifying returns epistasis, only differing in which genotype is considered the beginning of a mutant cycle.

To classify the types and frequencies of different forms of epistasis, we compared the estimated effect of each pairwise interaction to the estimated effect of each single amino acid involved in that pairwise interaction. The fraction of genetic variance explained by the particular pairwise effect was then added to the appropriate category, either sign, reciprocal sign, diminishing returns, or diminishing costs epistasis. To extend this approach to third-order interactions, we sequentially fixed each amino acid in the third order interaction, taking that genotype as the starting genotype and then varied the other two amino acids in a manner analogous to the pairwise epistasis calculation. The fraction of genetic variance explained by a particular third order interaction was then divided by three (as there are three states that can be fixed) and added to the appropriate category (*Figure 3*; *Figure 3—figure supplement 4*).

## Direction of the effect of mutations' effects

The genetic score of a specific genotype is the summation of the effects in that sequence, that is the single global term, four single site effects, six pairwise interactions, and four three-way interactions for both binding and specificity. The model thus gives the effect of each amino acid or interaction relative to the average of all genotypes, not the effect of individual mutations from one state to another. To calculate the effect of a mutation in a given genetic background, we calculated the difference in the genetic score between two genotypes with a single amino acid difference. This entails a change in one main effect, three pairwise interactions, and three third-order interactions. In this way, the model directly captures the fact that even a single amino acid change has the potential to alter numerous epistatic interactions. As a result, to determine the effect of a mutation requires measuring its impact across numerous genetic backgrounds. To determine the range of possible effects of a single mutation, we compared the genetic score between two genotypes differing by a single amino acid on each of the 8000 genetic backgrounds on which the particular change from one amino acid to another could occur and determined the frequency in which that particular mutation increased or decreased binding on a particular DNA element. To minimize the effect of small changes in genetic score upon mutation being counted, we only considered changes in genetic score greater than 5% of the difference between the average genotype and the threshold needed to be classified as a strong activator. We also used the fact that the model identifies which epistatic factors (main, pairwise, third-order) are responsible for the change in genetic score between variants, as well as how this effect is partitioned between binding and specificity, to identify when changes in epistasis were necessary, sufficient, or both for the change in genetic score (*Figure 4*; *Figure 4—figure supplement 1*).

## Sequence space analysis
### Network types

To determine how epistasis affected the evolution of DBD binding specificity, we generated connected networks for each model of epistasis using the R package igraph (*Csárdi and Nepusz, 2006*). In each case, genotypes were connected if they differed by a single amino acid (hamming distance) or if the single amino acid difference could occur via a single nucleotide mutation (genetic code distance). In the latter case, we accounted for the disjoint nature of serine codons in the genetic code, treating the two groups as non-interchangeable by coding all serine amino acids in both an S and a Z form. Thus, for a sequence containing n serine amino acid, there are $2^n$ versions of that amino acid sequence with the same phenotype that inhabit different locations in sequence space with different connectivity to neighboring sequences. We created networks for the full model, the model containing up to pairwise epistasis, and the model containing no specific epistasis (*Figure 5*; *Figure 5—figure supplement 1*). We used only those genotypes that were functional in a particular model, that is were either weak or strong on either ERE, SRE, or both. In addition, we created a network of all 160,000 genotypes. This

latter network contained no information about the function of each genotype and was used to model evolution over sequence space unconstrained by function.

## Network distance

Network distances were calculated as the shortest path distance between two genotypes, constrained to only take steps to neighboring genotypes in a network. Because only functional genotypes are present in a network, the distance between genotypes in a network can exceed four, even when connections are made by hamming distance, if there are no direct paths between the genotypes. We also calculated the distance from each ERE-specific genotype to the nearest SRE-specific genotype, of which there may be several, by finding the minimum distance for each ERE-specific genotype (*Figure 5—figure supplement 2*).

## Shared vs unique network genotypes

One of the main effects of epistasis is to influence whether a genotype is functional or not. Thus, many genotypes are only functional in one or a subset of networks. However, even if the same genotypes are functional with and without epistasis, epistasis can influence the paths between them that evolution is likely to take. To distinguish between these effects, we compared the effects of epistasis on path lengths for all functional genotypes and for the set of genotypes that are functional in all three networks.

## One-step functional transitions and epistatic necessity/sufficiency

To determine how epistasis influenced changing binding classes or specificity, we identified all instances in which two neighboring genotypes in a network differ in binding class or specificity. These represent portions of sequence space where a single amino acid mutation can move a genotype from one binding class to another or alter specificity from ERE-specific to SRE-specific. For changes between binding classes, we identified all such mutations on the full network with third order epistasis. For changes in specificity, we identified the mutation responsible for all such instances in the full network, as well as the networks that had no third-order epistasis or had no epistatic effects (*Figure 4*; *Figure 4—figure supplement 2*).

We calculated whether epistasis was necessary, sufficient, or both for transitions in binding classes and specificity. For binding classes, epistasis was categorized as necessary if the main effect of the amino acid change was not large enough to move the genotype into the new binding class without contribution from the epistatic effects. Similarly, epistasis was categorized as sufficient if epistatic changes were large enough to move the genotype into the new binding class without contribution from the main effects. For single step transitions in specificity, epistasis was necessary if the main, non-epistatic, effect of an amino acid change was not large enough to either reduce ERE binding or increase SRE binding enough to switch the binding class from ERE-specific to SRE-specific. Epistasis was sufficient if the epistatic effects alone were large enough to both reduce ERE binding and increase SRE binding enough to switch the binding class from ERE-specific to SRE-specific. In the full model, a similar distinction was made for third order epistasis relative to main effects and pairwise epistasis and calculated in an analogous manner.

## Number of paths and path distinctiveness

The number of paths between two genotypes was calculated as the number of distinct shortest paths between them. Two paths were considered distinct if at least one genotype along the paths were different. To account for the fact that many paths traverse similar sets of genotypes, we also developed a metric for distinctiveness of a set of paths between two genotypes. To calculate this metric, we extracted from the larger network the set of genotypes along any shortest path between the starting and ending genotype. Intuitively, for a path with S genotypes, the set of paths between the starting and ending genotypes should be considered more distinct if they contain more unique genotypes or fewer shared edges between intermediate genotypes. To capture these aspects, we calculated two metrics. The first metric relates the effective number of paths ($P_g$) to the number of unique genotypes (G) and the path length (S - 1). The second metric relates a different measure of the effective number of paths ($P_e$) to the number of unique edges (E) and the number of genotypes along a single path (S).

$$P_g = \frac{G-2}{S-1} \text{ and } P_e = \frac{E}{S}$$

The first calculation provides the maximum number of completely distinct paths with no shared genotypes that could be created given the path length and the number of genotypes present. It thus provides an upper bound on the number of distinct paths possible. If a set of paths are completely distinct, that is sharing no genotypes, then $P_g$ will equal $P_e$. However, if the paths are not completely distinct, then there will be more edges among the set of paths than if all paths were distinct. Thus, for a given number of genotypes and path length, the greater the number of edges connecting those genotypes, the less distinct the set of paths are. By comparing the difference in these metrics to the maximum possible value, we can estimate the effective number of distinct paths (D) in a set.

$$D = P_g - \left(P_e - P_g\right)\frac{P_g}{P_e} = \frac{P_g^2}{P_e} = \frac{S(\frac{G-2}{S-1})^2}{E}$$

We used this equation to calculate the distinctiveness of paths from each ERE-specific genotype to each SRE-specific genotype over the network.

## Evolutionary simulations
### Neutral model
To determine whether epistasis increased or decreased the ability of evolution to find a protein variant with altered specificity, we performed forward evolutionary simulations over the previously described genotype networks. These simulations captured neutral wandering among a set of functional variants and are similar to the verbal model of protein evolution described by *Smith, 1969*. In each simulation, evolution was constrained to move among only functional variants, that is those classified by the particular model as either ERE-specific, SRE-specific or promiscuous. Genotypes were classified as either ERE-specific if they bound ERE and not SRE, SRE-specific if they bound SRE and not ERE, and promiscuous if they bound both ERE and SRE. Genotypes were considered neighbors if a single nucleotide change could mutate one variant into the other (genetic code) or if they differed by a single amino acid (hamming distance). To estimate the global effect of epistasis, evolution was initiated from each ERE-specific genotype and allowed to take a random step to a neighboring genotype, regardless of DNA binding specificity, until an SRE-specific genotype was reached. This was repeated 100 times for each starting genotype. For the subset of ERE-specific genotypes found in each of the three models with varying amounts of epistasis (Full model, pairwise only, no epistasis), an additional 1000 simulations were performed to estimate how epistasis altered particular evolutionary paths. For each simulation, we recorded the number of steps, ending genotype identity, and the identity of each intermediate genotype during the evolutionary walk.

### Model with selection for increased specificity
To estimate how epistasis might interact with selection for increased specificity, we performed additional forward evolutionary simulations. As before, evolution was initiated from ERE-specific genotypes and terminated when an SRE-specific genotype was first encountered. Unlike the neutral simulations, however, we incorporated selection for increased specificity. At each step, the specificity of each neighboring genotype was calculated from the underlying genetic scores as the difference in binding to SRE and ERE. We then used Kimura's formula for fixation probability to calculate the relative probability of fixation among all possible neighbors (*Kimura, 1962*). The effects of specificity were scaled such that the difference between a non-binder and a strong binder corresponded to a scaled selection coefficient (i.e. Ns) of 1 in a population of size 100. We initiated evolution with a range of population sizes between 1 and 10,000 to capture the transition between neutrality and selection driven changes, performing 100 simulations for each population size on each network using the genetic code to connect genotypes. We noticed during these simulations, that for a small subset of starting genotypes, evolution would often become 'stuck' on local optimum in the pairwise model, continuously cycling between two genotypes. These genotypes were substantially more SRE-specific than the neighboring functional sequences. Simulations with strong selection for specificity (i.e. large population sizes) were therefore more likely to continuously cycle between these genotypes. To account for

this effect, we also estimated evolutionary path lengths after removing all evolutionary simulations in which more than 20 steps were taken (*Figure 5—figure supplement 3*).

## Acknowledgements

We thank members of the Thornton lab for helpful comments on the manuscript and the University of Chicago's Research Computing Center for computational resources.

## Additional information

### Funding

| Funder | Grant reference number | Author |
|---|---|---|
| National Institutes of Health | F32-GM122251 | Brian PH Metzger |
| National Institutes of Health | R01-GM131128 | Joseph W Thornton |
| National Institutes of Health | R01-GM121931 | Joseph W Thornton |
| National Institutes of Health | R01-GM139007 | Joseph W Thornton |
| Samsung Scholarship | | Yeonwoo Park |
| National Institutes of Health | R35-GM14533601 | Joseph W Thornton |

The funders had no role in study design, data collection and interpretation, or the decision to submit the work for publication.

### Author contributions

Brian PH Metzger, Software, Formal analysis, Validation, Investigation, Visualization, Methodology, Writing – original draft, Writing – review and editing; Yeonwoo Park, Methodology, Writing – review and editing; Tyler N Starr, Conceptualization, Methodology; Joseph W Thornton, Conceptualization, Funding acquisition, Project administration, Writing – review and editing

### Author ORCIDs

Brian PH Metzger ⓘ https://orcid.org/0000-0003-4878-2913
Joseph W Thornton ⓘ http://orcid.org/0000-0001-9589-6994

Reviewer #1 (Public Review): https://doi.org/10.7554/eLife.88737.3.sa1
Reviewer #2 (Public Review): https://doi.org/10.7554/eLife.88737.3.sa2
Author response https://doi.org/10.7554/eLife.88737.3.sa3

## Additional files

### Supplementary files
• MDAR checklist

### Data availability

The coding scripts for running all analyses are located on Github (https://github.com/JoeThorntonLab/DBD.GeneticARchitecture copy archived at *JoeThorntonLab, 2023*). Initial and intermediate data files can be found at dryad (https://doi.org/10.5061/dryad.jsxksn0hk).

The following dataset was generated:

| Author(s) | Year | Dataset title | Dataset URL | Database and Identifier |
|---|---|---|---|---|
| Metzger BPH, Park Y, Starr TN, Thornton JW | 2024 | Epistasis facilitates functional evolution in an ancient transcription factor | https://doi.org/10.5061/dryad.jsxksn0hk | Dryad Digital Repository, 10.5061/dryad.jsxksn0hk |

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

## Appendix 1

### Partitioning of Variance

In our modeling formalism, the importance of each model term – the fraction of total specific genetic variance that it accounts for – can be calculated directly from its magnitude and order. In addition, these microscopic contributions can be summed over any set of terms to yield the fraction of variance accounted for by the set. Specifically, the fraction of variance accounted for by any term is equal to the square of the term, weighted by the number of genotypes to which it applies, which is a function of the term's epistatic order (main, pairwise, third-order, etc.). This property flows ultimately from the constraint we placed on the model coefficients to sum to zero at each site and combination of sites. Here we show why this relationship holds.

We begin with the definition of specific genetic variance, which is the variance in the genetic score among all genotype:

$$Var\left(y\left(G\right)\right) = \frac{1}{2 \cdot 20^4} \sum_{g \in G} \left(y\left(g\right) - \overline{y\left(G\right)}\right)^2 \tag{A1}$$

Where g is a particular genotype on a particular DNA element, and y(g) the genetic score (phenotype) of that genotype, and $G$ the set of all possible protein sequences/RE element complexes. The variance is calculated over all genotypes for both ERE and SRE. The phenotype of a particular genotype y(g) can be expressed as the sum of the model coefficients for which that genotype has a particular amino acid state or combination of states. We retain a form of y(g) with both binding and specificity coefficients, although this distinction is not necessary for the proof. The binding coefficients ($\beta$) are the average effect on the genetic score of an amino acid or combination of amino acids, regardless of which DNA element is being considered. Specificity coefficients ($\sigma$) are one half of the difference in the genetic score between binding to ERE and SRE. In the convention we use, binding to ERE adds the specificity effects, and binding to SRE subtracts the specificity terms. Thus, for a particular genotype $g$ with sequence $q_1q_2q_3q_4$ we have:

$$y\left(g^E\right) = \beta_0 + \sum_i \beta_{q_i} + \sum_{i<j} \beta_{q_iq_j} + \sum_{i<j<k} \beta_{q_iq_jq_k} - \sigma_0 - \sum_i \sigma_{q_i} - \sum_{i<j} \sigma_{q_iq_j} - \sum_{i<j<k} \sigma_{q_iq_jq_k}$$

$$y\left(g^S\right) = \beta_0 + \sum_i \beta_{q_i} + \sum_{i<j} \beta_{q_iq_j} + \sum_{i<j<k} \beta_{q_iq_jq_k} - \sigma_0 - \sum_i \sigma_{q_i} - \sum_{i<j} \sigma_{q_iq_j} - \sum_{i<j<k} \sigma_{q_iq_jq_k} \tag{A2}$$

Here $g^E$ and $g^S$ are genotypes bound to ERE or SRE respectively, i,j, and k index the available sites, and $\beta_{q_i}/\sigma_{q_i}$ are the corresponding model terms based on the state of genotype $g$ at site i.

The global binding intercept $\beta_0$ is defined as the average phenotype of all genotypes on the two DNA elements:

$$\beta_0 = \frac{1}{2 \cdot 20^4} \sum_{g \in G} y\left(g\right) = \overline{y\left(G\right)} \tag{A3}$$

Similarly, the global specificity term $\sigma_0$ reflects the difference in the average phenotype of genotypes on one element relative to the global average, or one half of the difference in average phenotypic binding between the two DNA elements:

$$\sigma_0 = \frac{1}{20^4} \sum_{g \in G} y\left(g^E\right) - \beta_0 = \frac{1}{2 \cdot 20^4} \sum_{g \in G} y\left(g^E\right) - y\left(g^S\right) = \frac{1}{2} \left(\overline{y\left(G^E\right)} - \overline{y\left(G^S\right)}\right) \tag{A4}$$

The remaining model terms capture deviations from the expected phenotype in the absence of progressively higher order interactions:

$$\beta_{q_i} = \overline{y(G_{q_i})} - \beta_0$$

$$\beta_{q_iq_j} = \overline{y(G_{q_iq_j})} - (\beta_0 + \beta_{q_i} + \beta_{q_j})$$

$$\beta_{q_iq_jq_k} = \overline{y(G_{q_iq_jq_k})} - (\beta_0 + \beta_{q_i} + \beta_{q_j} + \beta_{q_k} + \beta_{q_iq_j} + \beta_{q_iq_k} + \beta_{q_jq_k})$$

$$\sigma_{q_i} = \frac{1}{2}\left(\overline{y(G_{q_i}^E)} - \overline{y(G_{q_i}^S)}\right) - \sigma_0 \tag{A5}$$

$$\sigma_{q_iq_j} = \frac{1}{2}\left(\overline{y(G_{q_iq_j}^E)} - \overline{y(G_{q_iq_j}^S)}\right) - (\sigma_0 + \sigma_{q_i} + \sigma_{q_j})$$

$$\sigma_{q_iq_jq_k} = \frac{1}{2}\left(\overline{y\left(G_{q_iq_jq_k}^E\right)} - \overline{y\left(G_{q_iq_jq_k}^S\right)}\right) - (\sigma_0 + \sigma_{q_i} + \sigma_{q_j} + \sigma_{q_k} + \sigma_{q_iq_j} + \sigma_{q_iq_k} + \sigma_{q_jq_k})$$

Where $G_{q_i}$ is the set of genotypes with state $q$ at site i. Because these terms capture deviations from an average, the set of terms at a site or combination of sites sum to zero by definition:

$$\sum_q \beta_{q_1} = \sum_q \beta_{q_2} = \sum_q \beta_{q_3} = \sum_q \beta_{q_4} = 0$$

$$\sum_{q^2} \beta_{q_1q_2} = \sum_{q^2} \beta_{q_1q_3} = \sum_{q^2} \beta_{q_1q_4} = \sum_{q^2} \beta_{q_2q_3} = \sum_{q^2} \beta_{q_2q_4} = \sum_{q^2} \beta_{q_3q_4} = 0$$

$$\sum_{q^3} \beta_{q_1q_2q_3} = \sum_{q^3} \beta_{q_1q_2q_4} = \sum_{q^3} \beta_{q_1q_3q_4} = \sum_{q^3} \beta_{q_2q_3q_4} = 0$$

$$\sum_q \sigma_{q_1} = \sum_q \sigma_{q_2} = \sum_q \sigma_{q_3} = \sum_q \sigma_{q_4} = 0 \tag{A6}$$

$$\sum_{q^2} \sigma_{q_1q_2} = \sum_{q^2} \sigma_{q_1q_3} = \sum_{q^2} \sigma_{q_1q_4} = \sum_{q^2} \sigma_{q_2q_3} = \sum_{q^2} \sigma_{q_2q_4} = \sum_{q^2} \sigma_{q_3q_4} = 0$$

$$\sum_{q^3} \sigma_{q_1q_2q_3} = \sum_{q^3} \sigma_{q_1q_2q_4} = \sum_{q^3} \sigma_{q_1q_3q_4} = \sum_{q^3} \sigma_{q_2q_3q_4} = 0$$

Where $q^2$ and $q^3$ are the set of all possible pairs or triplets of amino acid states respectively. Using the definition of $\beta_0$ and inserting **Equation A2** into **Equation A1**, we can rewrite the specific genetic variance as:

$$Var\left(y\left(G\right)\right) = \frac{1}{2 \cdot 20^4} \sum_{g \in G^E} \left(\sum_i \beta_{q_i} + \sum_{i<j} \beta_{q_iq_j} + \sum_{i<j<k} \beta_{q_iq_jq_k} + \sigma_0 + \sum_i \sigma_{q_i} + \sum_{i<j} \sigma_{q_iq_j} + \sum_{i<j<k} \sigma_{q_iq_jq_k}\right)^2$$

$$+ \frac{1}{2 \cdot 20^4} \sum_{g \in G^S} \left(\sum_i \beta_{q_i} + \sum_{i<j} \beta_{q_iq_j} + \sum_{i<j<k} \beta_{q_iq_jq_k} - \sigma_0 - \sum_i \sigma_{q_i} - \sum_{i<j} \sigma_{q_iq_j} - \sum_{i<j<k} \sigma_{q_iq_jq_k}\right)^2 \tag{A7}$$

As expected, the specific genetic variance is independent of the global binding coefficient $\beta_0$. However, it is not independent of the global specificity coefficient $\sigma_0$. We can rewrite **Equation A7** using a more compact notation as:

$$Var\left(y\left(G\right)\right) = \frac{1}{2 \cdot 20^4} \sum_{g \in G^E} \left(\sum_a \beta_a + \sigma_0 + \sum_a \sigma_a\right)^2 + \frac{1}{2 \cdot 20^4} \sum_{g \in G^S} \left(\sum_a \beta_a - \sigma_0 - \sum_a \sigma_a\right)^2 \tag{A8}$$

Where $a$ is the set of sites or combinations of sites for all orders other than the intercepts. Thus, $a$ includes all of the $q_i$ first order terms, the $q_iq_j$ second order terms, and the $q_iq_jq_k$ third order terms. We next expand the squared term, rewriting the sum as the sum of each squared coefficient plus the sum of the product of all pairwise combinations of coefficients:

$$Var\left(y\left(G\right)\right) = \frac{1}{2 \cdot 20^4} \sum_{g \in G^E} \left(\sum_a (\beta_a^2 + \sigma_a^2) + \sigma_0^2 + 2\sum_{a<b}(\beta_a\beta_b + \sigma_a\sigma_b + \beta_a\sigma_b) + 2\sum_a(\sigma_0\beta_a + \sigma_0\sigma_a)\right)$$

$$+ \frac{1}{2 \cdot 20^4} \sum_{g \in G^S} \left(\sum_a (\beta_a^2 + \sigma_a^2) + \sigma_0^2 + 2\sum_{a<b}(\beta_a\beta_b + \sigma_a\sigma_b - \beta_a\sigma_b) - 2\sum_a(\sigma_0\beta_a + \sigma_0\sigma_a)\right) \tag{A9}$$

Where b also indexes the sites/combinations of sites among the different epistatic orders. Distributing the outer sum over these terms gives:

$$Var\left(y\left(G\right)\right) = \frac{1}{2 \cdot 20^4} \left(\sum_a \sum_{g \in G^E} (\beta_a^2 + \sigma_a^2) + \sum_{g \in G^E} \sigma_a^2 + 2\sum_{a<b}\sum_{g \in G^E}(\beta_a\beta_b + \sigma_a\sigma_b + \beta_a\sigma_b) + 2\sum_a \sum_{g \in G^E}(\sigma_0\beta_a + \sigma_0\sigma_a)\right)$$

$$+ \frac{1}{2 \cdot 20^4} \left(\sum_a \sum_{g \in G^S} (\beta_a^2 + \sigma_a^2) + \sum_{g \in G^S} \sigma_a^2 + 2\sum_{a<b}\sum_{g \in G^S}(\beta_a\beta_b + \sigma_a\sigma_b - \beta_a\sigma_b) - 2\sum_a \sum_{g \in G^S}(\sigma_0\beta_a + \sigma_0\sigma_a)\right) \tag{A10}$$

To simplify this expression, we note that all of the cross-product terms in (10) turn out to be zero. To demonstrate this, we note that the set of all genotypes can be represented by conditioning on a particular amino acid state at a particular site and then taking the union over all possible amino acid states for that site:

$$G = \bigcup_{q_i} G_{q_i} \tag{A11}$$

We can thus reorganize a sum of cross product terms over all genotypes in (10) as a sum of cross product terms conditioned on a particular set of amino acid states and then sum over all possible amino acid states. For example, the sum of the cross-products of the binding coefficients with $a = q_1$ and $b = q_2$ can be rewritten by conditioning on the amino acid states at sites 1 and 2, and then summing over all possible combinations of amino acids for those sites:

$$\sum_{g \in G^E} \beta_{q_1} \beta_{q_2} = \sum_{q_1} \sum_{q_2} \sum_{g \in G_{q_1 q_2}^E} \beta_{q_1} \beta_{q_2} \tag{A12}$$

Where $G_{q_1 q_2}^E$ is the set of genotypes with state $q_1$ at site 1 and state $q_2$ at site 2 bound to ERE. From here, we note that once conditioned on a particular set of amino acid states at a set of sites, that combination of amino acid states will appear on all possible combinations of amino acids at the remaining sites. We can thus replace a sum over all genotypes with number of such combinations and then factor the result:

$$\sum_{q_1} \sum_{q_2} \sum_{g \in G_{q_1 q_2}^E} \beta_{q_1} \beta_{q_2} = \sum_{q_1} \sum_{q_2} 20^2 \cdot \beta_{q_1} \beta_{q_2} = 20^2 \cdot \sum_{q_1} \beta_{q_1} \sum_{q_2} \beta_{q_2} \tag{A13}$$

Since each of these latter sums is over the entire set of coefficients at a site, then by construction they sum to zero, and their product is thus also zero:

$$20^2 \cdot \sum_{q_1} \beta_{q_1} \sum_{q_2} \beta_{q_2} = 20^2 \cdot \sum_{q_1} \beta_{q_1} \cdot 0 = 0 \tag{A14}$$

A similar logic holds for higher order terms, for specificity terms, and for the combination of binding and specificity terms; as long as the conditional sums are over the entire set of amino acid combinations for those sites then we can replace summing over all genotypes with the number of genetic backgrounds a particular combination of states will appear on. For example, for the sum of cross products between binding and specificity terms with $a = q_1 q_2$ and $b = q_1 q_2 q_3$

$$\sum_{g \in G^E} \beta_{q_1 q_2} \sigma_{q_1 q_2 q_3} = \sum_{q_1} \sum_{q_2} \sum_{q_3} \sum_{g \in G_{q_1 q_2 q_3}^E} \beta_{q_1 q_2} \sigma_{q_1 q_2 q_3} = \sum_{q_1} \sum_{q_2} \sum_{q_3} 20 \cdot \beta_{q_1 q_2} \sigma_{q_1 q_2 q_3}$$
$$= 20 \cdot \sum_{q_1} \sum_{q_2} \beta_{q_1 q_2} \sum_{q_3} \sigma_{q_1 q_2 q_3} = 20 \cdot \sum_{q_1} \sum_{q_2} \beta_{q_1 q_2} \cdot 0 = 0 \tag{A15}$$

Similar logic can be applied to all terms in *Equation A10* that are the sum of the product of coefficients at other orders: the terms for any pair or triplet of sites always sum to zero, so the sum of the product of those terms (times any other set of terms) also sums to zero.

Combining *Equation A10* with this result, we can rewrite the specific genetic variance as:

$$Var(y(G)) = \frac{1}{2 \cdot 20^4} \left( \sum_{g \in G^E} \sigma_0^2 + \sum_a \sum_{g \in G^E} (\beta_a^2 + \sigma_a^2) \right) + \frac{1}{2 \cdot 20^4} \left( \sum_{g \in G^S} \sigma_0^2 + \sum_a \sum_{g \in G^S} (\beta_a^2 + \sigma_a^2) \right)$$
$$= \frac{1}{20^4} \left( \sum_{g \in G} \sigma_0^2 + \sum_a \sum_{g \in G} \beta_a^2 + \sigma_a^2 \right) \tag{A16}$$

Where the second equality follows because the sets of squared coefficients are the same for binding ERE and SRE. To simplify further, we again use the fact that the set of all genotypes can be partitioned into smaller sets based on the amino acid state at a site. For example, for the squared binding coefficients at site 1:

$$\frac{1}{20^4} \sum_{g \in G} \beta_{q_1}^2 = \frac{1}{20^4} \sum_{q_1} \sum_{g \in G_{q_1}} \beta_{q_1}^2 = \frac{1}{20^4} \sum_{q_1} 20^3 \cdot \beta_{q_1}^2 = \frac{1}{20} \sum_{q_1} \beta_{q_1}^2 \tag{A17}$$

Similarly, when coefficients involve multiple sites:

$$\frac{1}{20^4} \sum_{g \in G} \beta_{q_1 q_2}^2 = \frac{1}{20^4} \sum_{q_1} \sum_{q_2} \sum_{g \in G_{q_1 q_2}} \beta_{q_1 q_2}^2 = \frac{1}{20^4} \sum_{q_1} \sum_{q_2} 20^2 \cdot \beta_{q_1 q_2}^2 = \frac{1}{20^2} \sum_{q \in Q} \beta_{q_1 q_2}^2 \qquad (A18)$$

And the global specificity term:

$$\frac{1}{20^4} \sum_{g \in G} \sigma_0^2 = \frac{1}{20^4} \cdot 20^4 \cdot \sigma_0^2 = \sigma_0^2 \qquad (A19)$$

Extending this logic to all the squared terms and combining the results of *Equations A17–A19* with *Equation A16*, we get:

$$Var\left(y\left(G\right)\right) = \frac{1}{20} \sum_i \beta_{q_i}^2 + \frac{1}{20^2} \sum_{i<j} \beta_{q_i q_j}^2 + \frac{1}{20^3} \sum_{i<j<k} \beta_{q_i q_j q_k}^2 + \sigma_0^2 + \frac{1}{20} \sum_i \sigma_{q_i}^2 + \frac{1}{20^2} \sum_{i<j} \sigma_{q_i q_j}^2 + \frac{1}{20^3} \sum_{i<j<k} \sigma_{q_i q_j q_k}^2$$

$$(A20)$$

Which is the sum of the squared coefficients, each normalized by the number of terms at a particular epistatic order. We can rewrite this more compactly as:

$$Var\left(y\left(G\right)\right) = \sum_{\beta \neq \beta_0} \frac{\beta^2}{20^{O(\beta)}} + \sum_{\sigma} \frac{\sigma^2}{20^{O(\sigma)}} \qquad (A21)$$

where β and σ represent the individual binding and specificity terms, and $O$ is the epistatic order of each term, with $O=0$ for global (intercept) terms, 1 for main effects, 2 for pairwise interactions, and 3 for third order interactions. Said another way, the total variance is the sum of each term's squared effect, weighted by the number of genotypes that the term affects.

One consequence of the relationship between the effect size of a term, its order of epistasis, and the total specific genetic variance explained, is that the fraction of specific genetic variance explained by any single term is readily calculable. For example, the fraction of specific genetic variance explained by each first order binding term at a particular site is:

$$F(Var(\beta_{q_i})) = \frac{\frac{1}{20}\beta_{q_i}^2}{\sum_{\beta \neq \beta_0} \frac{\beta^2}{20^{O(\beta)}} + \sum_{\sigma} \frac{\sigma^2}{20^{O(\sigma)}}} = \frac{\frac{1}{20}\beta_{q_i}^2}{Var\left(y\left(G\right)\right)} \qquad (A22)$$

In addition, since the total specific genetic variance is the simple weighted sum of squared terms, the fraction of specific genetic variance explained by any set of terms is also easy to calculate. For example, the fraction of specific genetic variance explained by all first order binding terms at a particular site is:

$$F(Var(\Sigma_i \beta_{qi})) = \frac{\frac{\frac{1}{20}\sum_i \beta_{q_i}^2}{Var\left(y\left(G\right)\right)}}{\sum_{\beta \neq \beta_0} \frac{\beta^2}{20^{O(\beta)}} + \sum_{\sigma} \frac{\sigma^2}{20^{O(\sigma)}}} = \frac{\frac{1}{20}\sum_i \beta_{q_i}^2}{Var\left(y\left(G\right)\right)} \qquad (A23)$$

Thus, knowing only the value of a model coefficient and the epistatic order it pertains to, we can readily calculate the fraction of the total specific genetic variance contributed by an individual term

and thus any set of terms in the model, including across sites, epistatic orders, mechanisms of action, and the particular form of epistasis.

