## [Editor Report · eLife assessment]

This study includes **fundamental** findings on protein evolution, namely that changes in function are largely attributable to pairwise rather than higher-order interactions, and that epistasis potentiates rather than constrains evolutionary paths. **Compelling** evidence supporting the conclusions is provided by applying a new model to a previously generated experimental dataset on deep mutational scanning of the DNA-binding domain (DBD) of steroid hormone receptor. The implications of this work are of considerable interest to protein biochemistry, evolutionary biology, and numerous other fields.

---

## [Referee Report · Reviewer #1 (Public Review)]

Metzger et al develop a rigorous method filling an important unmet need in protein evolution - analysis of protein genetic architecture and evolution using data from combinatorially complete 20^N variant libraries. Addressing this need has become increasingly valuable, as experimental methods for generating these datasets expand in scope and scale. Their method integrates two key features - (1) it reports the effects of mutations relative to the average across all variants, rather than a particular genotype, making it useful for examining global genetic architecture, and (2) it does this for all possible 20 states at each site, in contrast to the binary analyses in prior work. These features are not individually novel but integrating them into a single analysis framework is novel and will be valuable to the protein evolution community. Using a previously published dataset generated by two of the authors, they conclude that (1) changes in function are largely attributable to pairwise but not higher-order interactions, and (2) epistasis potentiates, rather than constrains, evolutionary paths. These findings are well-supported by the data. Overall, this work has important implications for predicting the relationship between genotype and phenotype, which is of considerable interest to protein biochemistry, evolutionary biology, and numerous other fields.

---

## [Referee Report · Reviewer #2 (Public Review)]

The authors aimed to understand how epistasis influences the genetic architecture of the DNA-binding domain (DBD) of steroid hormone receptor. An ordinal regression model was developed in this study to analyze a published deep mutational scanning dataset that consists of all combinatorial amino acid variants across four positions (i.e. 160,000 variants). This published dataset measured the binding of each variant to the estrogen receptor response element (ERE, sequence: AGGTCA) as well as the steroid receptor response element (SRE, sequence: AGAACA). This model has major strengths of being reference free and able to account for global nonlinearity in the genotype-phenotype relationship. Thorough analyses of the modelling results have performed, which provided convincing results to support the importance of epistasis in promoting evolution of protein functions. This conclusion is impactful because many previous studies have shown that epistasis constrains evolution. The novelty this study will likely stimulate new ideas in the field. The model will also likely be utilized by other groups in the community.

---

## [Author Response]

The following is the authors’ response to the original reviews.

**Response to Reviewers’ Public Comments**

We are grateful for the reviewers’ comments. We have modified the manuscript accordingly and detail our responses to their major comments below.

(1) Reviewer 2 was concerned that transformation of continuous functional data into categorical form could reduce precision in estimating the genetic architecture.

We agree that transforming continuous data into categories may reduce resolution, but it also improves accuracy when the continuous data are affected by measurement noise. In our dataset, many genotypes are at the lower bound of measurement, and the variation in measured fluorescence among these genotypes is largely or entirely caused by measurement noise. By transforming to categorical data, we dramatically reduced the effect of this noise on the estimation of genetic effects. We modified the results and discussion sections to address this point.

(2) Reviewer 2 asked about generalizability of our findings.

Because our paper is the first use of reference-free analysis of a 20-state combinatorial dataset, generalizability is at this point unknown. However, a recent manuscript from our group confirms the generality of the simplicity of genetic architecture: using reference-free methods to analyze 20 published combinatorial deep mutational scans, several of which involve 20-state libraries, we found that main and pairwise effects account for virtually all of the genetic variance across a wide variety of protein families and types of biochemical functions (Park Y, Metzger BPH, Thornton JW. 2023. The simplicity of protein sequence-function relationships. BioRxiv, 2023.09.02.556057). Concerning the facilitating effect of epistasis on the evolution of new functions, we speculate that this result is likely to be general: we have no reason to think that the underlying cause of this observation – epistasis brings genotypes with different functions closer in sequence space to each other and expands the total number of functional sequences – arises from some peculiarity of the mechanisms of steroid receptor DBD folding or DNA binding. However, we acknowledge that our data involve sequence variation at those sites in the protein that directly mediate specific protein-DNA contact; it is plausible that sites far from the “active site” may have weaker epistatic interactions and therefore have weaker effects on navigability of the landscape. We have addressed these issues in the discussion.

(3) Reviewer 3 asked “in which situation would the authors expect that pairwise epistasis does *not* play a crucial role for mutational steps, trajectories, or space connectedness, if it is dominant in the genotype-phenotype landscape?”

The question addressed in our paper is not whether epistasis shapes steps, trajectories or connectedness in sequence space but how it does so and what its particular effects are on the evolution of new functions. The dominant view in the field has been that the primary role of epistasis is to block evolutionary paths. We show, however, that in multi-state sequence space, epistasis facilitates rather than impedes the evolution of new functions. It does this by increasing the number of functional genotypes and bringing genotypes with different functions closer together in sequence space. This finding was possible because of the difference in approach between our paper and prior work: most prior work considered only direct paths in a binary sequence space between two particular starting points – and typically only considering optimization of a single function – whereas we studied the evolution of new functions in a multi-state amino acid space, under empirically relevant epistasis informed by complete combinatorial experiments. The result is a clear demonstration that the net effect of real-world levels of epistasis on navigability of the multidimensional sequence landscape is to make the evolution of new functions easier, not harder.

(4) Reviewer 3 asked for “an explanation of how much new biological results this paper delivers as compared with the paper in which the data were originally published.”

Starr 2017 did not use their data to characterize the underlying genetic architecture of function by estimating main and epistatic effects of amino acid states and combinations; it also did not evaluate the importance of epistasis in generating functional variants, determining the transcription factor’s specificity, or shaping evolutionary navigability on the landscape.

(5) Reviewer 3 requested an explanation of how the results would have been (potentially) different if a reference-based approach were used, and how reference-based analysis compares with other reference-free approaches to estimating epistasis.

This topic has been covered in detail in a recent manuscript from our group (Park et al. Biorxiv 2023.09.02.556057). Briefly, reference-free approaches provide the most efficient explanation of an entire genotype-phenotype map, explaining the maximum amount of genetic variance and reducing sensitivity to experimental noise and missing genotypes compared to reference-based approaches. Reference-based approaches tend to infer much more epistasis, especially higher-order epistasis, because measurement error and local idiosyncrasy near the wild-type sequence propagate into spurious high-order terms. Reference-based analyses are appropriate for characterizing only the immediate sequence neighborhood of a particular “wild-type” protein of interest. Reference-free approaches are therefore best suited to understanding genotype-phenotype landscapes as a whole. We have clarified these issues in the revised discussion.

(6) Reviewer 3 suggested that the comparison between the full and main-effects-only model should involve a re-estimation of main effects in the latter case.

This is indeed what we did in our analysis. We have clarified the description in the results and methods sections to make this clear.

(7) Reviewer 3 asked about the applicability of the approach to data beyond those analyzed in the present study and requirements to use it.

Our approach could be used for any combinatorial DMS dataset in which the phenotypic data are categorical (or can be converted to categorical form). Complete sampling is not required: a virtue of reference-free analysis is that by averaging the estimated effects of states and combinations over all variants that contain them, reference-free analysis is highly robust to missing data (except at the highest possible order of epistasis, where only a single variant represents a high-order effect) as long as variant sampling is unbiased with respect to phenotype. All the required code are publicly available at the github link provided in this manuscript. We have also described a general form of reference-free analysis for continuous data and applied it to 20 protein datasets in a recent publication (Park et al. Biorxiv 2023.09.02.556057).

(8)Reviewer 3 suggested that the text could be shortened and made less dense.

We agree and have done a careful edit to streamline the narrative.

**Response to Reviewers’ Non-Public Recommendations**
(1) Reviewer 1 noted that specific epistatic effects might in some cases produce global nonlinearities in the genotype-phenotype relationship. They then asked how our results might change if we did not impose a nonlinear transformation as part of the genotype-phenotype model. The reviewer’s underlying concern was that the non-specific transformation might capture high-order specific epistatic effects and thus reducing their importance.

Because our data are categorical, we required a model that characterizes the effect of particular amino acid states and combinations on the probability that a variant is in a null, weak, or strong activation class. A logistic model is the classic approach to this kind of analysis. The model structure assumes that amino acid states and combinations have additive effects on the log-odds of being in one functional class versus the lower functional class(es); the only nonlinear transformation is that which arises mathematically when log-odds are transformed into probability through the logistic link function. Thinking through the reviewer’s comment, we have concluded that our model does not make any explicit transformation to account for nonlinearity in the relationship between the effects of specific sequence states/combinations and the measured phenotype (activation class). If additional global nonlinearities are present in the genotype-phenotype relationship – such as could be imposed by limited dynamic range in the production of the fluorescence phenotype or the assay used to measure it – it is possible that the sigmoid shape of the logistic link function may also accommodate these nonlinearities. We have noted this part in the revised manuscript.

(2) Reviewer 1 observed that our model seems to prefer sets of several pairwise interactions among states across sites rather than fewer high-order interactions among those same states.

This finding arises because the pattern of phenotypic variation across genotypes in our dataset is consistent with that which would be produced by pairwise interactions rather than by high-order interactions. In a reference-free framework, these patterns are distinct from each other: a group of second-order terms cannot fit the patterns produced by high-order epistasis, and high-order terms cannot fit the pattern produced by pairwise interactions. Similarly, main-effect terms cannot fit the pattern of phenotypes produced by a pairwise interaction, and a pairwise epistatic term cannot fit the pattern produced by main effects of states at two sites. For example, third-order terms are required when the genotypes possessing a particular triplet of states deviate from that expected given all the main and second-order effects of those states; this deviation cannot be explained by any combination of first- and second-order effects.

We explain this point in detail in our recent manuscript (Park Y, Metzger BPH, Thornton JW. 2023. The simplicity of protein sequence-function relationships. BioRxiv, 2023.09.02.556057) and we summarize it here. Consider the simple example of two sites with two possible states (genotypes 00, 01, 10, and 11). If there are no main effects and no pairwise effects, this architecture will generate the same phenotype for all four variants – the global average (or zero-order effect). If there are pairwise effects but no main effects, this architecture will generate a set of phenotypes on which the average phenotype of genotypes with a 0 at the first site (00 and 01) equals the global average – as does the average of those with 0 at the second site (00 and 10). The epistatic effect causes the individual genotypes to deviate from the global average. This pattern can be fit only by a pairwise epistatic term, not by first-order terms. Conversely, if there are main effects but no pairwise effects, then the average phenotype of genotypes 00 and 01 will deviate from the global average (by an amount equal to the first-order effect), as will the average of (00 and 10): the phenotype of each genotype will be equal to the sum of the relevant first-order effects for the state it contains. This pattern cannot be fit by second-order model terms. The same logic extends to higher orders: a cluster of second-order terms cannot explain variation generated by third-order epistasis, because third-order variation is by definition is the deviation from the best second-order model.

(3) Reviewer 1 suggested several places in the text where citations to prior work would be appropriate.

We appreciate these suggestions and have modified the manuscript to refer to most of these works.

(4) Reviewer 1 pointed to the paper of Gong et al eLife 2013 and asked whether it is known how robust the proteins in our study are to changes in conformation/stability compared to other proteins, and whether this might impact the likelihood of observing higher-order epistasis in this system.

The DBDs that we study here are very stable, and previous work shows that mutations affect DNA specificity primarily by modifying the DBD’s affinity rather than its stability (McKeown et al., Cell 2014). Additionally, Gong et al.’s findings pertain to a globally nonlinear relationship between stability and function, which arises from the Boltzmann relationship between the energy of folding and occupancy of the folded state. Because our data are categorical – based on rank-order of measured phenotype rather than fluorescence as a continuous phenotype – the kind of global nonlinearity observed in Gong’s study are not expected to produce spurious estimates of epistasis in our work. We have modified the discussion to discuss the point.

(5) Reviewer 1 asked (a) why the epistatic models produce landscapes on which variants have fewer neighbors on average than main-effects only models and (b) why the average distance from all ERE-specific nodes to all SRE-specific nodes is greater with epistasis (but the average distance from ERE to nearest SRE is lower with epistasis).

In the main effects-only landscape, the functional genotypes are relatively similar to each other, because each must contain several of the states that contribute the most to a positive genetic score. Moreover, ERE-specific nodes are similar to each other, and SRE-specific nodes are similar to each other, because each must contain one or more of a relatively small number of specificity-determining states. When epistasis is added to the genetic architecture, two things happen: (1) more genotypes become functional because there are more combinations that can exceed the threshold score to produce a functional activator and (2) these additional functional variants are more different from each other – in general, and within the classes of ERE- or SRE-specific variants – because there are now more diverse combinations of states that can yield either phenotype. As a result, a broader span of sequence space is occupied, but ERE- and SRE-specific variants are more interspersed with each other. This means that the average distance between all pairs of nodes is greater, and this applies to all ERE-SRE pairs, as well. However, the interspersing means that the closest single SRE to any particular ERE is closer than it was without epistasis. We have added this explanation to the main text.

(6) Reviewer 2 asked us to explain why average path length increases with pairwise epistasis as the strength of selection for specificity increases.

This behavior occurs because of the existence of a local peak in the pairwise model. Genotypes on this peak contained few connections to other genotypes, all of which were less SRE specific. Thus, with strong selection, i.e. high population size, the simulations became stuck on the local peak, cycling among the genotypes many times before leaving, resulting in a large increase in the mean step number. As shown in the rest of the figure, when the longest set of paths are removed, there are still differences in the average number of steps with and without epistasis. This issue is described in the methods section.

(7) Reviewers made several suggestions for clarity in the text and figures.

We have modified the paper to address all of these comments.

(8) Reviewer 3 stated that the code should be available.

The code is available at https://github.com/JoeThorntonLab/DBD.GeneticArchitecture.